# FEWER MAY BE BETTER: ENHANCING OFFLINE REINFORCEMENT LEARNING WITH REDUCED DATASET

**Yiqin Yang**[1], **Quanwei Wang**[2], **Chenghao Li**[2], **Hao Hu**[2], **Chengjie Wu**[2], **Yuhua Jiang**[2],
**Dianyu Zhong**[2], **Ziyou Zhang**[2], **Qianchuan Zhao**[2], **Chongjie Zhang**[3], **Bo Xu**[1†]
[1]The Key Laboratory of Cognition and Decision Intelligence for Complex Systems,
 Institute of Automation, Chinese Academy of Sciences
[2]Tsinghua University
[3]Washington University in St. Louis
`yiqin.yang@ia.ac.cn`

## ABSTRACT

Offline reinforcement learning (RL) represents a significant shift in RL research, allowing agents to learn from pre-collected datasets without further interaction with the environment. A key, yet underexplored, challenge in offline RL is selecting an optimal subset of the offline dataset that enhances both algorithm performance and training efficiency. Reducing dataset size can also reveal the minimal data requirements necessary for solving similar problems. In response to this challenge, we introduce ReDOR (Reduced Datasets for Offline RL), a method that frames dataset selection as a gradient approximation optimization problem. We demonstrate that the widely used actor-critic framework in RL can be reformulated as a submodular optimization objective, enabling efficient subset selection. To achieve this, we adapt orthogonal matching pursuit (OMP), incorporating several novel modifications tailored for offline RL. Our experimental results show that the data subsets identified by ReDOR not only boost algorithm performance but also do so with significantly lower computational complexity.

## 1 INTRODUCTION

Offline reinforcement learning (RL) (Levine et al., 2020) has marked a paradigm shift in artificial intelligence. Unlike traditional RL (Sutton & Barto, 2018) that relies on real-time interaction with the environment, offline RL utilizes pre-collected datasets to learn decision-making policies (Yang et al., 2021; Janner et al., 2021). This approach is increasingly favored for its practicality in scenarios where real-time data acquisition is impractical or could damage physical assets. Moreover, offline learning can avoid the significant time and complexity involved in online sampling and environment construction. This streamlines the learning process and expands the potential for deploying RL across a more comprehensive array of applications (Yuan et al., 2022; Zhou et al., 2023; Nambiar et al., 2023).

However, offline reinforcement learning relies on large pre-collected datasets, which can result in substantial computational costs during policy learning (Lu et al., 2022), especially when the algorithm model requires extensive parameter tuning (Sharir et al., 2020). Moreover, additional data may not always improve performance, as suboptimal data can exacerbate the distribution shift problem, potentially degrading the policy (Hu et al., 2022). In this work, we attempt to explore effective offline reinforcement learning methods through a data subset selection mechanism and address the following question:

*How do we determine the subset of the offline dataset to improve algorithm performance and accelerate algorithm training?*

In this paper, we formulate the dataset selection challenge as a gradient approximation optimization problem. The underlying rationale is that if the weighted gradients of the TD loss on the reduced

---

[†]Corresponding Author

dataset can closely approximate those on the original dataset, the dataset reduction process should not lead to significant performance degradation. However, directly solving this data selection problem is NP-Hard (Killamsetty et al., 2021b;c). To this end, we first prove that the common actor-critic framework can be transformed into a submodular optimization problem (Mirzasoleiman et al., 2020). Based on this insight, we adopt the Orthogonal Matching Pursuit (OMP) (Elenberg et al., 2018) to solve the data selection problem. On the other hand, different from supervised learning, target values in offline RL evolve with policy updates, resulting in unstable gradients that affect the quality of the selected data subset. To solve this issue, we stabilize the learning process by making several essential modifications to the OMP.

Theoretically, we provide a comprehensive analysis of the convergence properties of our algorithm and establish an approximation bound for its solutions. We then prove the objective function can be upper-bounded if the selected data is sufficiently diverse. Empirically, we evaluate REDOR on the D4RL benchmark (Fu et al., 2020). Comparison against various baselines and ablations shows that the data subsets constructed by the REDOR can significantly improve algorithm performance with low computationally expensive. To the best of our knowledge, our work is the first study analyzing the reduced dataset in offline reinforcement learning.

## 2 RELATED WORKS

**Offline Reinforcement Learning.** Current offline RL methods attempted to constrain the learned policy and behavior policy by limiting the action difference (Fujimoto et al., 2019), adding KL-divergence (Yang et al., 2021; Nair et al., 2020; Peng et al., 2019; Wu et al., 2019), regularization (Kumar et al., 2019), conservative estimates (Yang et al., 2023; Ma et al., 2021a; Kumar et al., 2020; Ma et al., 2021b) or penalizing uncertain actions (Janner et al., 2019; Yu et al., 2021; Kidambi et al., 2020). These studies provide a solid foundation for implementing and transferring reinforcement learning to real-world tasks.

**Offline Dataset.** Some works attempted to explore which dataset characteristics dominate in offline RL algorithms (Schweighofer et al., 2021; Swazinna et al., 2021; Chen et al., 2020; Yue et al., 2022) or investigate the data generation (Yarats et al., 2022). Recently, some researchers attempted to solve the sub-optimal trajectories issue by constraining policy to good data rather than all actions in the dataset (Hong et al., 2023b) or re-weighting policy (Hong et al., 2023a). However, limited research has addressed considerations related to the reduced dataset in offline RL.

**Data Subset Selection.** The research on identifying crucial samples within datasets is concentrated on supervised learning. Some prior works use uncertainty of samples (Coleman et al., 2019; Paul et al., 2021) or the frequency of being forgotten (Toneva et al., 2018) as the proxy function to prune the dataset. Another research line focuses on constructing weighted data subsets to approximate the full dataset (Feldman, 2020), which often transforms the subset selecting to the submodular set cover problem (Wei et al., 2015; Kaushal et al., 2019). These studies establish the importance of selecting critical samples from datasets for practical training. However, unlike supervised learning, target values in offline RL evolve as policies update, leading to unstable gradients that significantly complicate the learning process.

## 3 BACKGROUND

**Reinforcement Learning (RL)** deals with Markov Decision Processes (MDPs). A MDP can be modeled by a tuple $(S, A, r, p, \gamma)$, with the state space $S$, the action space $A$, the reward function $r(s, a)$, the transition function $p(s'|s, a)$, and the discount factor $\gamma$. We follow the common assumption that the reward function is positive and bounded: $\forall s \in S, a \in A, 0 \leq r(s, a) \leq R_{\max}$, where $R_{\max}$ is the maximum possible reward. RL aims to find a policy $\pi(a \mid s)$ that maximizes the cumulative discounted return:

$$\pi^* = \arg\max_\pi J(\pi) = \arg\max_\pi \mathbb{E}_\pi[\sum_{t=0}^{H} \gamma^t r(s_t, a_t)], \tag{1}$$

where $H$ is the horizon length. For any policy $\pi$, the action value function is $Q^\pi(s_t, a_t) = \mathbb{E}_\pi[\sum_{k=0}^{H-t} \gamma^k r(s_{t+k}, a_{t+k})|s_t{=}s, a_t{=}a]$. The state value function is $V^\pi(s_t) =$

$\mathbb{E}_\pi[\sum_{k=0}^{H-t} \gamma^k r(s_{t+k}, a_{t+k})|s_t{=}s]$. It follows from the Bellman equation that $V^\pi(s_t) = \sum_{a \in A} \pi(a|s)Q^\pi(s_t, a_t)$.

**Offline RL** learns a policy $\pi$ without interacting with an environment. Rather, the learning is based on a dataset $\mathcal{D}$ generated by a behavior policy $\pi_\beta$. One of the major challenges in offline RL is the issue of distributional shift (Fujimoto et al., 2019), where the learned policy is different from the behavioral policy. Existing offline RL methods apply various forms of regularization to limit the deviation of the current learned policy:

$$\pi^* = \arg\max_\pi \left[ J_\mathcal{D}(\pi) - \alpha D(\pi, \pi_\beta) \right], \tag{2}$$

where $J_\mathcal{D}(\pi)$ is the cumulative discounted return of policy $\pi$ on the empirical MDP induced by the dataset $\mathcal{D}$, and $D(\pi, \pi_\beta)$ is a divergence measure between $\pi$ and $\pi_\beta$. In this paper, we base our study on TD3+BC (Fujimoto & Gu, 2021), which follows this regularized learning scheme.

We introduce the concept of **offline data subset selection**. Specifically, let $\mathcal{D} = \{(s_i, a_i, r_i, s'_i)\}_{i=1}^M$ denote the complete offline dataset, and let $\mathcal{S} \subseteq \mathcal{D}$, indexed by $j$, represent the reduced dataset. We formulate the subset selection as:

$$\mathcal{S}^* = \arg\min_{\mathcal{S} \subseteq \mathcal{D}} |\mathcal{S}|, \quad \text{s.t.} \quad J(\pi_\mathcal{S}) \geq J(\pi_\mathcal{D}) + c, \tag{3}$$

where $\pi_\mathcal{D}$ and $\pi_\mathcal{S}$ are the policy trained using Eq. 2 with dataset $\mathcal{D}$ and $\mathcal{S}$, respectively. $c \geq 0$ is the policy performance gain.

**Compact Subset Selection** for offline reinforcement learning remains largely under-explored in existing literature. However, research efforts have been directed toward reducing the size of training samples in other deep learning fields like supervised learning (Killamsetty et al., 2021a;b; Mirzasoleiman et al., 2020).

Specifically, there are some research explorations on transforming the subset selection problem into the submodular set cover problem (Mirzasoleiman et al., 2020). The submodular set cover problem is defined as finding the smallest set $\mathcal{S}$ that achieves utility $\rho$:

$$\mathcal{S}^* = \arg\min_{\mathcal{S} \subseteq \mathcal{D}} |\mathcal{S}|, \quad \text{s.t.} \quad F(\mathcal{S}) \geq \rho, \tag{4}$$

where we slightly abuse the notation and use $\mathcal{D}$ to denote the complete supervised learning dataset. We require $F$ to be a *submodular* function like set cover and concave cover modular (Iyer et al., 2021). A function $F$ is submodular if it satisfies the *diminishing returns property*: for subsets $\mathcal{S} \subseteq \mathcal{T} \subseteq \mathcal{D}$ and $j \in \mathcal{D} \setminus \mathcal{T}$, $F(j \mid \mathcal{S}) \triangleq F(\mathcal{S} \cup j) - F(\mathcal{S}) \geq F(j \mid \mathcal{T})$ and the *monotone property*: $F(j \mid \mathcal{S}) \geq 0$ for any $j \in \mathcal{D} \setminus \mathcal{S}$ and $\mathcal{S} \subseteq \mathcal{D}$.

## 4 METHOD

For the data subset selection problem, RL and supervised learning are significantly different in two aspects: (1) In supervised learning, the loss value is the primary criterion for selecting data. However, the loss value in RL is unrelated to the policy performance. Therefore, we need to consider new criteria for selecting data in RL. (2) Compared with the fixed learning objective in supervised learning, the learning objective in offline RL evolves as policies update, significantly complicating the data selection process. To solve these issues, we first formulate the data selection problem in offline RL as the constrained optimization problem in Sec. 4.1. Then, we present how to effectively solve the optimization problem in Sec. 4.2. Finally, we balance the data quantity with performance in Sec. 4.3. The algorithm framework is shown in Algorithm 1.

### 4.1 GRADIENT APPROXIMATION OPTIMIZATION

We first approximate the optimization problem 3, using the Q-function $Q^\pi(s, a)$ as the performance measure $J(\pi) = Q^\pi(s_0, a_0)$ and requiring that $Q^{\pi_\mathcal{D}}$ and $Q^{\pi_\mathcal{S}}$ to be approximately equal for any action-state pair $(s, a)$:

$$\mathcal{S}^* = \arg\min_{\mathcal{S} \subseteq \mathcal{D}} |\mathcal{S}|, \quad \text{s.t.} \quad \|Q^{\pi_\mathcal{D}}(s, a) - Q^{\pi_\mathcal{S}}(s, a)\|_\infty \leq \delta. \tag{5}$$

We use *gradient approximation optimization* to deal with the constraint in the optimization problem 5. Suppose that Q-functions are represented by networks with learnable parameters $\theta$ and updated by gradients of loss function $\mathcal{L}(\theta)$, e.g., the TD loss (Mnih et al., 2015). If we can identify a reduced training set $\mathcal{S}$ such that the weighted sum of the gradients of its elements closely approximates the full gradient over the complete dataset $\mathcal{D}$, then we can train on $\mathcal{S}$ and converge to a Q-function that is nearly identical to the one trained on $\mathcal{D}$.

Formally,

$$\mathcal{L}(\theta) = \sum_{i \in \mathcal{D}} \mathcal{L}^i(\theta) = \sum_{i \in \mathcal{D}} \mathcal{L}_{\text{TD}}(s_i, a_i, r_i, s_i', \theta) \tag{6}$$

is the standard Q-learning TD loss, and

$$\mathcal{L}_{\text{rdc}}(\boldsymbol{w}, \theta) = \sum_{i \in \mathcal{S}} w_i \mathcal{L}^i(\theta) \tag{7}$$

is the loss on the reduced subset $\mathcal{S} \subseteq \mathcal{D}$. In order to better approximate the gradient for the full dataset, we use the weighted data subset. Specifically, $w_i$ is the per-element weight in coreset $\mathcal{S}$. During the learning process, we approximate the entire dataset's gradient by multiplying the samples' gradient in coreset by their weights. We define the following error term:

$$\text{Err}\,(\boldsymbol{w}, \mathcal{S}, \mathcal{L}, \theta) = \|\sum_{i \in \mathcal{S}} w_i \nabla_\theta \mathcal{L}^i(\theta) - \nabla_\theta \mathcal{L}(\theta)\|_2. \tag{8}$$

Minimizing Eq. 8 ensures the dataset selection procedure can maintain or even improve the policy performance. Similarly, define the regularized version of $\text{Err}\,(\boldsymbol{w}, \mathcal{S}, \mathcal{L}, \theta)$ as

$$\text{Err}_\lambda\,(\boldsymbol{w}, \mathcal{S}, \mathcal{L}, \theta) = \text{Err}\,(\boldsymbol{w}, \mathcal{S}, \mathcal{L}, \theta) + \lambda \|\boldsymbol{w}\|_2^2. \tag{9}$$

Then, the optimization problem 3 is transformed into:

$$\boldsymbol{w}, \mathcal{S} = \underset{\boldsymbol{w}, \mathcal{S}}{\arg\min}\,\text{Err}_\lambda\,(\boldsymbol{w}, \mathcal{S}, \mathcal{L}, \theta). \tag{10}$$

## 4.2 Orthogonal Matching Pursuit for Offline RL

Directly solving problem 10 is NP-hard (Killamsetty et al., 2021b;c) and computationally intractable. To solve the issue, we consider using the iterative approach, which selects data one by one to reduce $\text{Err}_\lambda\,(\boldsymbol{w}, \mathcal{S}, \mathcal{L}, \theta)$. To ensure newly selected data are informative, we prove the optimized problem 10 can be transformed into the submodular function.

Specifically, we introduce a constant $L_{\max}$ and define $F_\lambda(\mathcal{S}) = L_{\max} - \min_{\boldsymbol{w}} \text{Err}_\lambda\,(\boldsymbol{w}, \mathcal{S}, \mathcal{L}, \theta)$. Then, we consider the common actor-critic framework in data subset selection, which has an actor-network $\pi_\phi(s)$ and a critic network $Q_\theta(s, a)$ that influence the TD loss and thus the function $F_\lambda(\mathcal{S})$. Therefore, the submodularity analysis of $F_\lambda(\mathcal{S})$ involves two components: $F_\lambda^Q(\mathcal{S})$ that depends on the critic loss $\mathcal{L}_Q(\theta)$, and $F_\lambda^\pi(\mathcal{S})$ that depends on the actor loss $\mathcal{L}_\pi(\phi)$. The following theorem shows that both $F_\lambda^Q(\mathcal{S})$ and $F_\lambda^\pi(\mathcal{S})$ are weakly submodular.

**Theorem 4.1** (Submodular Objective). *For $|\mathcal{S}| \leq N$ and sample $(s_i, a_i, r_i, s_i') \in \mathcal{D}$, suppose that the TD loss and gradients are bounded: $|\mathcal{L}^i(\theta)| \leq U_{\text{TD}}$, $\|\nabla_\theta Q_\theta(s_i, a_i)\|_2 \leq U_{\nabla Q}$, $\|\nabla_{\pi_\phi(s_i)} Q_\theta(s_i, \pi_\phi(s_i))\|_2 \leq U_{\nabla a}$, $\|\pi_\phi(s_i) - a_i\|_2 \leq U_a$, $\|\pi_\phi(s_i)\|_2 \leq U_\pi$, and $\|\nabla_\phi \pi_\phi(s_i)\|_2 \leq U_{\nabla \pi}$, then $F_\lambda^Q(\mathcal{S})$ is $\delta$-weakly submodular, with*

$$\delta \geq \frac{\lambda}{\lambda + 4N(U_{\text{TD}} U_{\nabla Q})^2}, \tag{11}$$

*and $F_\lambda^\pi(\mathcal{S})$ is $\delta$-weakly submodular, with*

$$\delta \geq \frac{\lambda}{\lambda + N(U_{\nabla a}/\alpha + 2U_a U_\pi)^2 U_{\nabla \pi}^2}. \tag{12}$$

Please refer to Appendix A.2 for detailed proof.

Based on the above theoretical analysis, let $\mathrm{Err}_\lambda\left(\boldsymbol{w}, \mathcal{S}_{j-1}, \mathcal{L}, \theta\right)$ represent the residual error at iteration $j$. Then, we adopt the Orthogonal Matching Pursuit (OMP) algorithm (Elenberg et al., 2018), which selects a new data sample $i$ and takes its gradient $\nabla_\theta \mathcal{L}^i(\theta)$ as the new basis vector to minimize this residual error. In this way, we update the residual to $\mathrm{Err}_\lambda\left(\boldsymbol{w}, \mathcal{S}_j, \mathcal{L}, \theta\right)$, where $\mathcal{S}_j = \mathcal{S}_{j-1} \cup \{i\}$. However, the dynamic nature of offline RL poses a challenge when using OMP, leading to unstable learning. To address the unique challenges of offline RL, we propose the following novel techniques to enhance gradient matching:

**(I) Stabilizing Learning with Changing Targets.** In supervised learning, the stability of training targets leads to stable gradients. However, in offline RL, target values evolve with policy updates, resulting in unstable gradients in Eq. 8 that affect the quality of the selected data subset. To address this issue, we will stabilize the learning process by using **empirical returns from trajectories** to smooth the gradient updates. This provides a more consistent learning signal and mitigates instability caused by changing target values. Specifically, rather than adopt the gradient of the TD loss, we calculate gradient $\nabla_\theta \mathcal{L}(\theta)$ from the following equation

$$\nabla_\theta \mathcal{L}(\theta) = \nabla_\theta \mathbb{E}_\mathcal{D}[(y - Q_\theta(s_t, a_t))^2], \quad y = \sum_{k=0}^{H-t} \gamma^k r(s_{t+k}, a_{t+k}). \tag{13}$$

Furthermore, we will adopt a **multi-round selection strategy** where data selection occurs over multiple rounds $T$. In each round, a portion of the data is selected based on the updated Q-values, reducing variance and ensuring that the subset captures the most critical information. This multi-round approach allows for dynamic adjustment of the selected subset as learning progresses, improving stability and reducing the risk of overfitting to specific trajectories. Specifically, we calculate $\nabla_{\theta_t} \mathcal{L}(\theta_t)$ at each round based on Eq. 13, where $\theta_t$ is the parameter updated in the $t$-round. In practice, we pre-store parameters $\theta_t$ with various rounds $t$ and load them during training.

**(II) Trajectory-based Selection.** In offline RL, collected data is often stored in trajectories, which are coherent and more valuable than individual data points. For this reason, we modify OMP to the trajectory-based gradient matching. Specifically, we select a new trajectory $i$ of length $K$ and take the mean of gradients $\nabla_{\theta_t} \mathcal{L}^i_{\mathrm{Traj}}(\theta_t) = \sum_{k=1}^{K} \nabla_{\theta_t} \mathcal{L}^k(\theta_t) / K$ as the new basis vector to minimize the residual error. Then, we update the residual to $\mathrm{Err}_\lambda\left(\boldsymbol{w}, \mathcal{S}_j, \mathcal{L}, \theta\right)$, where $\mathcal{S}_j = \mathcal{S}_{j-1} \cup \{\mathrm{Trajectory}_i\}$.

### 4.3 Balancing Data Quantity with Performance

In offline RL, while additional data can help generalization, suboptimal data may lead to significant performance degradation due to distribution shifts. To address this, we will introduce a **constraint term** that biases the TD-gradient matching method toward selecting data with higher return estimates. Then, based on the design in the Sec. 4.2 (I), the Equation 13 is transformed into

$$\nabla_\theta \mathcal{L}(\theta) = \nabla_\theta \mathbb{E}_\mathcal{D}[(y - Q_\theta(s_t, a_t))^2], \quad y = \sum_{k=0}^{H-t} \gamma^k r(s_{t+k}, a_{t+k}),$$
$$\text{s.t.} \quad y > \mathrm{Top}\, m\%(\{\mathrm{Return}(\mathrm{Trajectory}_j)\}_{j=1}^{|\mathcal{D}|}). \tag{14}$$

This regularized constraint selection approach ensures that the selected subset not only reduces computational costs but also focuses on data points that are aligned with the learned policy, avoiding performance degradation caused by suboptimal trajectories.

## 5 Theoretical Analysis

In this section, we study the convergence property of our method and the error bounds of the solutions it finds. We work with mild assumptions that the gradient of the TD loss is Lipschitz smooth with constant $L$: $\|\nabla \mathcal{L}(\theta') - \nabla \mathcal{L}(\theta)\| \leq L\|\theta' - \theta\|$, and that the gradient is bounded by $\sigma$: $\|\nabla \mathcal{L}(\theta)\| \leq \sigma$.

Firstly, we show that the TD loss of the offline Q function $Q^{\pi_\mathcal{S}}$ trained on the reduced dataset $\mathcal{S}$ can converge.

---

**Algorithm 1** Reduce Dataset for Offline RL (REDOR)

---

1: **Require**: Complete offline dataset $\mathcal{D}$
2: Initialize parameters of the offline agent for data selection $Q_\theta, \pi_\phi$
3: **for** $t = 1, \cdots, T$ **do**
4:     Load parameter $\theta_t$ for $Q_{\theta_t}$
5:     Calculate $\nabla_{\theta_t}\mathcal{L}(\theta_t), \nabla_{\theta_t}\mathcal{L}_{\text{Traj}}(\theta_t)$ based on Equation 14
6:     $\mathcal{S}_t, \boldsymbol{w}_t = \text{OMP}(\nabla_{\theta_t}\mathcal{L}(\theta_t), \nabla_{\theta_t}\mathcal{L}_{\text{Traj}}(\theta_t), \theta_t)$
7: **end for**
8: Reduced offline dataset $\mathcal{S} \leftarrow \cup_{t \in [T]}\mathcal{S}_t$
9: Initialize parameters of the offline agent for training on the reduced offline dataset $Q_\vartheta, \pi_\varphi$
10: Train $Q_\vartheta, \pi_\varphi$ based on $\mathcal{S}$ and $\boldsymbol{w}$

---

**Algorithm 2** OMP algorithm

---

1: **Require**: $\nabla_{\theta_t}\mathcal{L}(\theta_t), \nabla_{\theta_t}\mathcal{L}_{\text{Traj}}(\theta_t), \theta_t$, regularization coefficient $\lambda$
2: $r \leftarrow \text{Err}_\lambda(\boldsymbol{w}_t, \mathcal{S}_t, \mathcal{L}, \theta_t)$
3: **while** $r \geq \epsilon$ **do**
4:     $e = \arg\max_{i \notin \mathcal{S}_t} |\langle \nabla_{\theta_t}\mathcal{L}_{\text{Traj}}^i(\theta_t), r \rangle|$
5:     $\mathcal{S}_t \leftarrow \mathcal{S}_t \cup \{\text{Trajectory}_e\}$
6:     $\boldsymbol{w}_t \leftarrow \arg\min_{\boldsymbol{w}_t} \text{Err}_\lambda(\boldsymbol{w}_t, \mathcal{S}_t, \mathcal{L}, \theta_t)$
7:     $r \leftarrow \text{Err}_\lambda(\boldsymbol{w}_t, \mathcal{S}_t, \mathcal{L}, \theta_t)$
8: **end while**
9: **Return** $\mathcal{S}_t$ and $\boldsymbol{w}_t$

---

**Theorem 5.1.** *Let $\theta^*$ denote the optimal $Q^{\pi_\mathcal{S}}$ parameters, $\theta_t$ the parameters after $t$ training steps. We have*

$$\min_{t=1:G} \mathcal{L}(\theta_t) \leq \mathcal{L}(\theta^*) + \frac{D\sigma}{\sqrt{G}} + \frac{D}{G}\sum_{t=1}^{G-1}\varepsilon. \tag{15}$$

*Here $\mathcal{L}(\theta) = \sum_{i \in \mathcal{D}} \mathcal{L}_{\text{TD}}(s_i, a_i, r_i, s_i', \theta)$ is the TD loss, $G$ is the number of total training steps, $D = \|\theta^* - \theta_t\|$, and $\varepsilon = \text{Err}(\boldsymbol{w}, \mathcal{S}, \mathcal{L}, \theta_t)$ is the gradient approximation errors.*

*Proof.* Please refer to Appendix A.4 for detailed proof. □

We assume the gradients of selected data are diverse and they can be divided into $K$ clusters $\{\mathcal{C}_1, \cdots, \mathcal{C}_K\}$ with the cluster centers set $\mathcal{C} = \{c_1, \cdots, c_K\}$. Then, we prove the residual error $\text{Err}(\boldsymbol{w}, \mathcal{S}, \mathcal{L}, \theta)$ can be upper bounded:

**Theorem 5.2.** *The residual error $\text{Err}(\boldsymbol{w}, \mathcal{S}, \mathcal{L}, \theta)$ is upper bounded according to the sample's gradient of TD loss:*

$$\min_{\mathcal{C}} \sum_{i \in \mathcal{D}} \min_{c \in \mathcal{C}} \|\nabla_\theta \mathcal{L}^i(\theta) - \nabla_\theta \mathcal{L}^c(\theta)\|_2. \tag{16}$$

*Proof.* Please refer to Appendix A.3 for detailed proof. □

We then prove that the reduced dataset selected by our method can achieve a good approximation for the gradient calculated on the complete dataset, which also means $\varepsilon = \text{Err}(\boldsymbol{w}, \mathcal{S}, \mathcal{L}, \theta_t)$ in Theorem 5.1 is bounded.

**Corollary 5.3** (Approximation Error Bound of the Reduced Dataset)**.** *The expected gradient approximation error achieved by our method is at most $5(\ln K + 2)$ times the error of the optimal solution $\mathcal{S}^*$:*

$$\text{Err}(\boldsymbol{w}, \mathcal{S}, \mathcal{L}, \theta) \leq 5(\ln K + 2)\text{Err}(\boldsymbol{w}, \mathcal{S}^*, \mathcal{L}, \theta). \tag{17}$$

*Proof.* The proof is derived by applying Theorem 5.2 along with Theorem 4.3 from (Makarychev et al., 2020), by observing that cluster centers are included in the reduced dataset. □

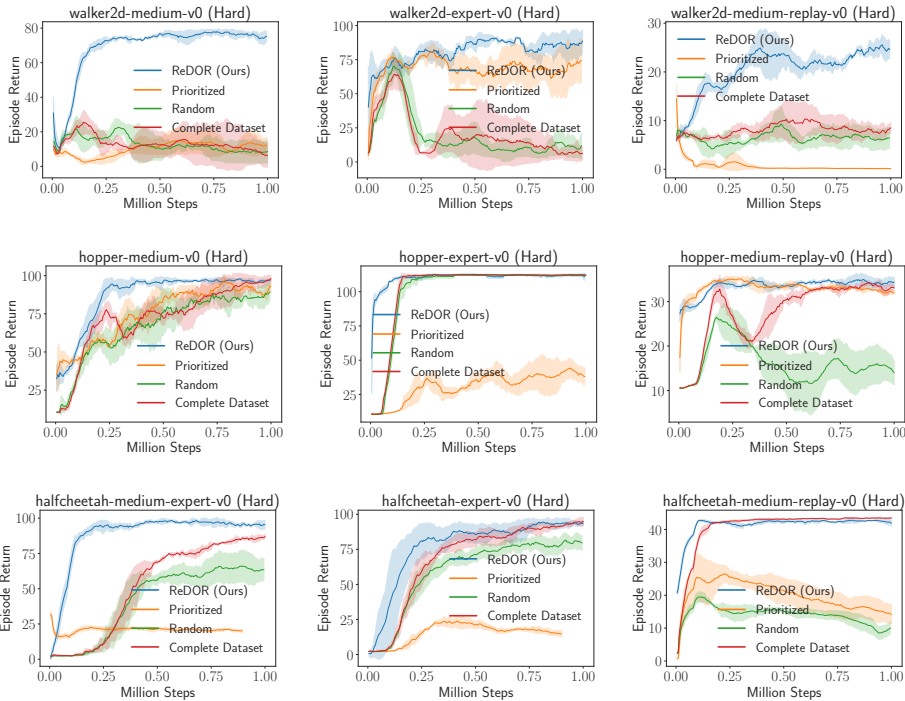

Figure 1: Experimental results on the D4RL (Hard) offline datasets. All experiment results were averaged over five random seeds. Our method achieves better or comparable results than the baselines with lower computational complexity.

**Discussion**    The aforementioned theoretical analysis has the following limitations: First, we assume that the gradients are uniformly bounded. Therefore, if the gradients of the algorithm diverge in practice, it would contradict our assumptions, and the selected data subset would no longer be valuable. Current offline RL methods can only ensure that the Q-values do not diverge Kumar et al. (2020); Fujimoto & Gu (2021). Although this can, to some extent, reflect the gradients of the Q-network that have not diverged, there is no rigorous proof that the bounds of the gradients can be guaranteed. Second, the above theoretical analysis is based on the classic TD loss. However, to provide a consistent learning signal and mitigate instability caused by changing target value, the techniques in Section 4.3 adopt a fixed target rather than TD loss.

# 6    EXPERIMENT

In this section, we assess the efficacy of our algorithm by addressing the following key questions. (1) Can offline RL algorithms achieve stronger performance on the reduced datasets selected by REDOR? (2) How does REDOR perform compare to other offline data selection methods? (3) What are the factors that contribute to REDOR's effectiveness?

## 6.1    SETUP

We evaluate algorithms on the offline RL benchmark D4RL (Fu et al., 2020) to answer the afore-mentioned questions. In addition, we consider a more challenging scenario where we add additional low-quality data to the dataset to simulate noise in real-world tasks, named D4RL (hard). The evaluation process commences with the selection of offline data, followed by the training of a widely recognized offline RL algorithm, TD3+BC (Fujimoto & Gu, 2021), on this reduced dataset for 1 million time steps. To ensure a fair comparison, we apply the same offline RL algorithm to data subsets obtained by different algorithms. Evaluation points are set at every 5,000 training time steps and involve calculating the return of 10 episodes per point. The results, comprising averages and standard deviations, are computed with five independent random seeds. On the other hand, we can also incorporate our method into offline model-based approaches, such as MOPO (Yu et al., 2020)

|  | KRLS | Log-Det | BlockGreedy | REDOR |
|---|---|---|---|---|
| Hopper-medium-v0 | 69.4±2.5 | 58.4±3.6 | 83.7±2.2 | **94.3±4.6** |
| Hopper-expert-v0 | 91.0±1.1 | 90.7±1.3 | 98.7±0.5 | **110.0±0.5** |
| Hopper-medium-replay-v0 | 28.5±3.2 | 29.4±1.2 | 30.5±2.4 | **35.3±3.2** |
| Walker2d-medium-v0 | 49.1±2.8 | 47.5±3.4 | 53.3±3.6 | **80.5±2.9** |
| Walker2d-expert-v0 | 68.4±3.2 | 67.5±5.6 | 74.8±3.4 | **104.6±2.5** |
| Walker2d-medium-replay-v0 | 14.3±1.2 | 15.2±2.2 | 16.7±1.3 | **21.1±1.8** |
| Halfcheetah-medium-v0 | 23.4±0.5 | 21.9±0.9 | 27.5±0.7 | **41.0±0.2** |
| Halfcheetah-expert-v0 | 73.9±1.4 | 72.1±2.2 | 79.2±1.8 | **88.5±2.4** |
| Halfcheetah-medium-replay-v0 | 39.5±0.3 | 39.9±0.5 | 40.5±1.0 | **41.1±0.4** |

Table 1: Experimental results on the D4RL (Hard) offline datasets. All experiment results were averaged over five random seeds. Our method performs better than the dataset reduction baselines.

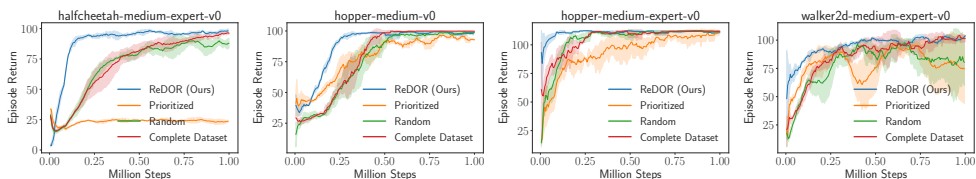

Figure 2: Experimental results on the D4RL offline datasets. All experiment results were averaged over five random seeds. Our method achieves better or comparable results than the baselines consistently.

and MoERL (Kidambi et al., 2020). Similarly, we only need to replace the current offline loss with the corresponding policy and model loss.

**Baselines**. We compare REDOR with data selection methods in RL. Specifically, previous work on prioritized experience replay for online RL (Schaul et al., 2015) aligns closely with our objective. We make this a baseline `Prioritized` where samples with the highest TD losses form the reduced dataset. Baseline `Complete Dataset` presents the performance by training TD3+BC with the original, complete dataset. Baseline `Random` randomly selects subsets from the D4RL dataset that are of the same size as REDOR. We also compare our method with general dataset reduction techniques from supervised learning. Specifically, we adopt the coherence criterion from Kernel recursive least squares (`KRLS`) (Engel et al., 2004), the log det criterion by forward selection in informative vector machines (`LogDet`) (Seeger, 2004) and the adapting kernel representation (`BlockGreedy`) (Schlegel et al., 2017) as our baselines.

## 6.2 EXPERIMENTAL RESULTS

**Answer of Question 1:** To show that REDOR can improve offline RL algorithms, we compare REDOR with Complete Dataset, Prioritized, and Random in the Mujoco domain. The experimental results in Figure 1 show that our method achieves superior performance than baselines. By leveraging the reduced dataset generated from REDOR, the agent can learn much faster than learning from the complete dataset. Further, the results in Figure 2 show that REDOR also performs better than the complete dataset and data selection RL baselines in the standard D4RL datasets. This is because prior methods select data in a random or loss-priority manner, which lacks guidance for subset selection and leads to degraded performance for downstream tasks.

In addition, to test REDOR's generality across various offline RL algorithms on various domains, we also conduct experiments on Antmaze tasks. We use IQL (Kostrikov et al., 2021) as the backbone of offline RL algorithms. The experimental results in Table 2 show that our method achieves stronger performance than baselines. In the antmaze tasks, the agent is required to stitch together various trajectories to reach the target location. In this scenario, randomly removing data could result in the loss of critical data, thereby preventing complete the task. Differently, REDOR extracts valuable

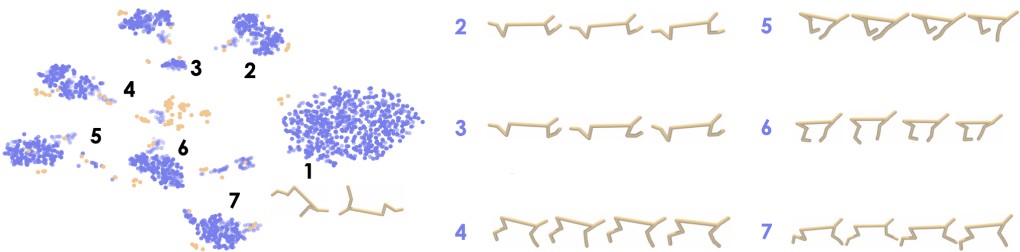

Figure 3: Visualization of the complete dataset and the reduced dataset in `halfcheetah` task. The higher opacity of a point represents a large time step towards the end of an episode. The dataset embedding is characterized by its division into different components. Samples selected by REDOR connect different components by focusing on the data related to the task.

| Env | Random | Prioritized | Complete Dataset | REDOR |
|---|---|---|---|---|
| Antmaze-umaze-v0 | 75.1±2.5 | 70.2±3.6 | 87.5±1.3 | **90.7±3.3** |
| Antmaze-umaze-diverse-v0 | 46.3±1.9 | 44.7±2.7 | 62.2±2.0 | **76.7±2.2** |
| Antmaze-medium-play-v0 | 59.3±1.6 | 60.3±2.9 | 71.2±2.2 | **80.3±2.9** |
| Antmaze-medium-diverse-v0 | 43.6±2.7 | 46.9±3.8 | 70.0±1.6 | **84.9±3.8** |
| Antmaze-large-play-v0 | 3.7±0.7 | 15.0±3.5 | 39.6±3.6 | **46.0±3.5** |
| Antmaze-large-diverse-v0 | 16.0±3.6 | 20.5±3.7 | 47.5±1.1 | **52.0±3.7** |

Table 2: Experimental results on the Antmaze offline datasets. All experiment results were averaged over five random seeds. Our method performs better than baselines.

subset by balancing data quantity with performance, achieving a stronger performance than the complete dataset.

**Answer of Question 2:** To test whether REDOR can select more valuable data than the data selection algorithms in supervised learning, we compare our method with KRLS (Engel et al., 2004), Log-Det (Seeger, 2004) and BlockGreedy (Schlegel et al., 2017) in the D4RL (Hard) datasets. The experimental results in Table 1 show that our method generally outperforms baselines. We hypothesize that supervised learning is static with fixed learning objectives, while offline RL's dynamic nature makes the target values evolve with policy updates, complicating the data selection process. Therefore, the data selection methods in supervised learning cannot be directly applied to offline RL scenarios.

**Answer of Question 3:** To study the contribution of each component in our learning framework, we conduct the following ablation study. `Q Target`: We replace the empirical returns used to update Q functions with the standard target Q function in the TD loss function. `Single Round`: We set the number of data selection rounds to 1 and study the function of multi-round data selection. The experimental results in Figure 4 in Appendix B show that removing any of these two modules will worsen the performance of REDOR. In case like `walker2d-medium`, ablation `Single Round` even decrease the performance by over 80%, and ablation `Q Target` results in a 95% performance drop in `walker2d-expert`. Furthermore, we also find that in the `halfcheetah` tasks, the impact of removing the two modules is relatively small. This result can be attributable to the fact that this task has a limited state space, and we can directly apply OMP to the entire dataset and identify important and diverse data.

We visualize the selected data by REDOR to better understand how it works. Figure 3 displays the t-SNE low-dimensional embeddings, with the complete dataset in blue and the selected data in orange. The higher opacity of a point indicates a larger time step. The dataset's structure is revealed by its segmentation into diverse components: In `halfcheetah`, each component reflects a distinct skill of the agent. For example, from 1 to 7, they represent falling, leg lifting, jumping, landing, leg swapping, stepping, and starting, respectively. We can observe that the selected samples by REDOR not only cover each component of the dataset but also effectively bridge the gaps between them, enhancing the dataset's versatility and coherence. Moreover, we find that REDOR is less concerned with the falling

data and instead focuses on the data related to the task. This observation can explain the improved performance of REDOR. For additional visualizations, please refer to Appendix C.1.

### 6.3 COMPUTATIONAL COMPLEXITY

We report the computational overhead of REDOR on various datasets. All experiments are conducted on the same computational device (GeForce RTX 3090 GPU). The results in Appendix C indicate that even on datasets containing millions of data points, the computational overhead of our method remains low (e.g., several minutes). This low computational complexity can be attributed to the trajectory-based selection technique in Sec. 4.2 (II) and the regularized constraint technique in Sec. 4.3, making our method easily scalable to large-scale datasets.

## 7 CONCLUSION

In this work, we demonstrate a critical problem in offline RL – identifying the reduced dataset to improve offline algorithm performance with low computational complexity. We cast the issue as the gradient approximation problem. By transforming the common actor-critic framework into the submodular objective, we apply the orthogonal matching pursuit method to construct the reduced dataset. Further, we propose multiple key modifications to stabilize the learning process. We validate the effectiveness of our proposed data selection method through theoretical analysis and extensive experiments. For future work, we attempt to apply our method to robot tasks in the real world.

## ACKNOWLEDGMENTS

This work was supported by Strategic Priority Research Program of the Chinese Academy of Sciences under Grant No.XDA27040200, in part by the National Key R&D Program of China under Grant No.2022ZD0116405.

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

# A PROOFS OF THEORETICAL ANALYSIS

## A.1 NOTATIONS

| Notation | Explanation |
|---|---|
| $U_{\text{TD}}$ | Bound of TD Loss |
| $U_{\nabla Q}$ | Bound of Gradient |
| $U_{\nabla a}$ | Bound of Gradient |
| $U_a$ | Bound of Action Difference |
| $U_\pi$ | Bound of Action |
| $\mathcal{D}$ | Complete Dataset |
| $\mathcal{S}$ | Reduced Dataset |
| $N$ | Size of Reduced Dataset |
| $\lambda$ | Minimum Eigenvalues |
| $\mathcal{C}$ | Cluster |
| $G$ | Total Training Steps |
| $\epsilon$ | Gradient Approximation Errors |
| $\theta_t$ | Updated parameter at the $t^{th}$ epoch |
| $\theta_t^*$ | Optimal model parameter |

Table 3: Organization of the notations used througout this paper

## A.2 SUBMODULAR

**Theorem 4.1** (Submodular Objective). *For $|\mathcal{S}| \leq N$ and sample $(s_i, a_i, r_i, s_i') \in \mathcal{D}$, suppose that the TD loss and gradients are bounded: $|\mathcal{L}^i(\theta)| \leq U_{\text{TD}}$, $\|\nabla_\theta Q_\theta(s_i, a_i)\|_2 \leq U_{\nabla Q}$, $\|\nabla_{\pi_\phi(s_i)} Q_\theta(s_i, \pi_\phi(s_i))\|_2 \leq U_{\nabla a}$, $\|\pi_\phi(s_i) - a_i\|_2 \leq U_a$, $\|\pi_\phi(s_i)\|_2 \leq U_\pi$, and $\|\nabla_\phi \pi_\phi(s_i)\|_2 \leq U_{\nabla \pi}$, then $F_\lambda^Q(\mathcal{S})$ is $\delta$-weakly submodular, with*

$$\delta \geq \frac{\lambda}{\lambda + 4N(U_{\text{TD}} U_{\nabla Q})^2}, \tag{11}$$

*and $F_\lambda^\pi(\mathcal{S})$ is $\delta$-weakly submodular, with*

$$\delta \geq \frac{\lambda}{\lambda + N(U_{\nabla a}/\alpha + 2U_a U_\pi)^2 U_{\nabla \pi}^2}. \tag{12}$$

*Proof.* As mentioned in Section 3, we use the TD3+BC algorithm as the basic offline RL algorithm. TD3+BC follows the actor-critic framework, which trains policy and value networks separately. For a single sample $(s_i, a_i, r_i, s_i')$, the loss of the value network is also named as TD error, which is defined by:

$$\mathcal{L}_Q^i(\theta) = (y_i - Q_\theta(s_i, a_i))^2 \tag{18}$$

$$\text{where} \quad y_i = r_i + \gamma Q_{\theta'}(s_i', \pi_{\phi'}(s_i') + \epsilon) \tag{19}$$

$$\tag{20}$$

The gradient is:

$$-\frac{1}{2} \nabla_\theta \mathcal{L}_Q^i(\theta) = (y_i - Q_\theta(s_i, a_i)) \nabla_\theta Q_\theta(s_i, a_i) \tag{21}$$

Offline RL algorithms attempt to minimize the TD error and compute the Q-value through a neural network. Therefore, we assume the upper bound of the TD error is $\max_i \|y_i - Q_\theta(s_i, a_i)\|_2 \leq U_{\text{TD}}$. The upper bound of the gradient of the value network is $\max_i \|\nabla_\theta Q_\theta(s_i, a_i)\|_2 \leq U_{\nabla Q}$. Then, Equation 21 can be transformed into:

$$\|\nabla_\theta \mathcal{L}_Q^i(\theta)\|_2 \leq 2U_{\text{TD}} U_{\nabla Q} \tag{22}$$

Similarly, for a single sample $(s_i, a_i, r_i, s'_i)$, the loss of the policy network is

$$\mathcal{L}_\pi^i(\phi) = -\frac{1}{\alpha} Q_\theta(s_i, \pi_\phi(s_i)) + \|\pi_\phi(s_i) - a_i\|_2^2 \tag{23}$$

$$\tag{24}$$

The gradient is:

$$\nabla_\phi \mathcal{L}_\pi^i(\phi) = \frac{\partial \mathcal{L}_\pi^i(\phi)}{\partial \pi_\phi(s_i)} \times \frac{\partial \pi_\phi(s_i)}{\partial \phi} \tag{25}$$

$$= [-\frac{1}{\alpha} \nabla_{\pi_\phi(s_i)} Q_\theta(s_i, \pi_\phi(s_i)) + 2(\pi_\phi(s_i) - a_i)^\top \pi_\phi(s_i)] \times \nabla_\phi \pi_\phi(s_i) \tag{26}$$

Here $\alpha$ is used to balance the conservatism and generalization in Offline RL, which is defined by:

$$\alpha = \frac{\mathbb{E}_{(s_i, a_i)}[|Q(s_i, a_i)|]}{\kappa} \tag{27}$$

where $\kappa$ is a hyper-parameter in TD3+BC. Note that although $\alpha$ includes $Q$, it is not differentiated over.

Offline RL algorithms attempt to limit the deviation of the current learned policy from the behavior policy while maximizing the Q-value of the optimized policy. Therefore, we assume the upper bound of the gradient of the value network is $\max_i \|\nabla_{\pi_\phi(s_i)} Q_\theta(s_i, \pi_\phi(s_i))\|_2 \leq U_{\nabla a}$. The upper bound of the action error is $\max_i \|\pi_\phi(s_i) - a_i\|_2 \leq U_a$. The upper bound of the output of the policy is $\max_i \|\pi_\phi(s_i)\|_2 \leq U_\pi$. The upper bound of the gradient of the policy network is $\max_i \|\nabla_\phi \pi_\phi(s_i)\|_2 \leq U_{\nabla \pi}$.

Then, Equation 26 can be bound:

$$\|\nabla_\phi \mathcal{L}_\pi^i(\phi)\|_2 \leq (U_{\nabla a}/\alpha + 2U_a U_\pi) U_{\nabla \pi} \tag{28}$$

We can define two functions $l_Q(\beta), l_\pi(\beta) : \mathbb{R}^{|\mathcal{D}|} \to \mathbb{R}$

$$l_Q(\beta) = -\|\sum_{i=1}^{|\mathcal{D}|} \beta_i \nabla_\theta \mathcal{L}_Q^i(\theta) - \nabla_\theta \mathcal{L}(\theta)\|_2 - \lambda \|\beta\|_2^2$$

$$\tag{29}$$

$$l_\pi(\beta) = -\|\sum_{i=1}^{|\mathcal{D}|} \beta_i \nabla_\phi \mathcal{L}_\pi^i(\phi) - \nabla_\phi \mathcal{L}(\phi)\|_2 - \lambda \|\beta\|_2^2$$

We assume $\beta$ is a $N$-sparse vector that is 0 on all but $N$ indices. Then we can transform maximizing $F_\lambda^Q(\mathcal{S}), F_\lambda^\pi(\mathcal{S})$ into maximizing $l(\beta) - l(\mathbf{0})$:

$$\max_{\mathcal{S}:|\mathcal{S}| \leq N} F_\lambda^Q(\mathcal{S}) \leftrightarrow \max_{\substack{\beta:\beta_{S^c}=0 \\ |\mathcal{S}| \leq N}} l_Q(\beta) - l_Q(\mathbf{0})$$

$$\tag{30}$$

$$\max_{\mathcal{S}:|\mathcal{S}| \leq N} F_\lambda^\pi(\mathcal{S}) \leftrightarrow \max_{\substack{\beta:\beta_{S^c}=0 \\ |\mathcal{S}| \leq N}} l_\pi(\beta) - l_\pi(\mathbf{0})$$

where $S^c$ means the complementary set of $S$, and $\beta_{S^c} = 0$ means $\beta$ is 0 on all but indices $i$ that $i \in S$. $l(\mathbf{0})$ means the value of $l(\cdot)$ when input is zero vector $\mathbf{0}$, it serves as a basic value. Since $l_Q(\beta) \leq 0, l_\pi(\beta) \leq 0$, we can easily find that the minimum eigenvalues of $-l_Q(\beta)$ and $-l_\pi(\beta)$ are both at least $\lambda$.

Next, the maximum eigenvalues of $-l_Q(\beta)$ and $-l_\pi(\beta)$ are

$$\Lambda_{\max}(-l_Q(\beta)) = \lambda + \mathrm{Trace}\left(\begin{bmatrix} \beta_1 \nabla_\theta \mathcal{L}_Q^{1\top}(\theta) \\ \beta_2 \nabla_\theta \mathcal{L}_Q^{2\top}(\theta) \\ \cdots \\ \beta_{|\mathcal{D}|} \nabla_\theta \mathcal{L}_Q^{|\mathcal{D}|\top}(\theta_t) \end{bmatrix} \begin{bmatrix} \beta_1 \nabla_\theta \mathcal{L}_Q^{1\top}(\theta) \\ \beta_2 \nabla_\theta \mathcal{L}_Q^{2\top}(\theta) \\ \cdots \\ \beta_{|\mathcal{D}|} \nabla_\theta \mathcal{L}_Q^{|\mathcal{D}|\top}(\theta) \end{bmatrix}^\top \right)$$

$$= \lambda + \sum_{i=1}^{|\mathcal{D}|} \beta_i^2 \|\nabla_\theta \mathcal{L}_Q^i(\theta)\|^2$$

$$\leq \lambda + 4N(U_{\mathrm{TD}} U_{\nabla Q})^2$$

$$\Lambda_{\max}(-l_\pi(\beta)) \leq \lambda + N(U_{\nabla a}/\alpha + 2U_a U_\pi)^2 U_{\nabla \pi}^2 \tag{31}$$

Following the Theorem 1 in Elenberg et al. (2018), we can derive that $F_\lambda^Q(\mathcal{S})$ is $\delta$-weakly submodular with $\delta \geq \frac{\lambda}{\lambda + 4N(U_{\mathrm{TD}} U_{\nabla Q})^2}$. And $F_\lambda^\pi(\mathcal{S})$ is $\delta$-weakly submodular with $\delta \geq \frac{\lambda}{\lambda + N(U_{\nabla a}/\alpha + 2U_a U_\pi)^2 U_{\nabla \pi}^2}$.

$\square$

### A.3 UPPER BOUND OF RESIDUAL ERROR

**Theorem 5.2.** *The residual error* $\mathrm{Err}(\boldsymbol{w}, \mathcal{S}, \mathcal{L}, \theta)$ *is upper bounded according to the sample's gradient of TD loss:*

$$\min_{\mathcal{C}} \sum_{i \in \mathcal{D}} \min_{c \in \mathcal{C}} \|\nabla_\theta \mathcal{L}^i(\theta) - \nabla_\theta \mathcal{L}^c(\theta)\|_2. \tag{16}$$

*Proof.* The residual error is no larger than the special case where all $w_i$ are $|\mathcal{D}|/|\mathcal{S}|$:

$$\mathrm{Err}(\boldsymbol{w}, \mathcal{S}, \mathcal{L}, \theta) \leq \|\frac{|\mathcal{D}|}{|\mathcal{S}|} \sum_{i \in \mathcal{S}} \nabla_\theta \mathcal{L}^i(\theta) - \sum_{i \in \mathcal{D}} \nabla_\theta \mathcal{L}^i(\theta)\|_2.$$

Using Jensen's inequality, we have

$$\mathrm{Err}(\boldsymbol{w}, \mathcal{S}, \mathcal{L}, \theta) \leq \sum_{i \in \mathcal{D}} \|\nabla_\theta \mathcal{L}^i(\theta) - \frac{1}{|\mathcal{S}|} \sum_{s \in \mathcal{S}} \nabla_\theta \mathcal{L}^s(\theta)\|_2.$$

According to the monotone property of submodular functions, adding more samples to $S^k$ reduces the residual error. We assume $S^k$ starts with the cluster center $\{c_k\}$, it follows that

$$\mathrm{Err}(\boldsymbol{w}, \mathcal{S}, \mathcal{L}, \theta) \leq \sum_{i \in \mathcal{D}} \|\nabla_\theta \mathcal{L}^i(\theta) - \nabla_\theta \mathcal{L}^{c_k}(\theta)\|_2$$

$$= \sum_{i \in \mathcal{D}} \min_{c \in \mathcal{C}} \|\nabla_\theta \mathcal{L}^i(\theta) - \nabla_\theta \mathcal{L}^c(\theta)\|_2. \tag{32}$$

Eq. 32 is exactly the optimization objective typical of the clustering problem. $\square$

### A.4 CONVERGENCE ANALYSIS

**Theorem 5.1.** *Let* $\theta^*$ *denote the optimal* $Q^{\pi_\mathcal{S}}$ *parameters,* $\theta_t$ *the parameters after* $t$ *training steps. We have*

$$\min_{t=1:G} \mathcal{L}(\theta_t) \leq \mathcal{L}(\theta^*) + \frac{D\sigma}{\sqrt{G}} + \frac{D}{G} \sum_{t=1}^{G-1} \varepsilon. \tag{15}$$

*Here* $\mathcal{L}(\theta) = \sum_{i \in \mathcal{D}} \mathcal{L}_{\mathrm{TD}}(s_i, a_i, r_i, s_i', \theta)$ *is the TD loss,* $G$ *is the number of total training steps,* $D = \|\theta^* - \theta_t\|$, *and* $\varepsilon = \mathrm{Err}(\boldsymbol{w}, \mathcal{S}, \mathcal{L}, \theta_t)$ *is the gradient approximation errors.*

*Proof.* From the definition of Gradient Descent, we have:

$$\nabla_\theta \mathcal{L}_{\mathrm{rdc}}(\theta_t)^T(\theta_t - \theta^*) = \frac{1}{\alpha_t}(\theta_t - \theta_{t+1})^T(\theta_t - \theta^*) \tag{33}$$

$$\nabla_\theta \mathcal{L}_{\mathrm{rdc}}(\theta_t)^T(\theta_t - \theta^*) = \frac{1}{2\alpha_t}\left(\|\theta_t - \theta_{t+1}\|^2 + \|\theta_t - \theta^*\|^2 - \|\theta_{t+1} - \theta^*\|^2\right) \tag{34}$$

$$\nabla_\theta \mathcal{L}_{\mathrm{rdc}}(\theta_t)^T(\theta_t - \theta^*) = \frac{1}{2\alpha_t}\left(\|\alpha_t \nabla_\theta \mathcal{L}_{\mathrm{rdc}}(\theta_t)\|^2 + \|\theta_t - \theta^*\|^2 - \|\theta_{t+1} - \theta^*\|^2\right) \tag{35}$$

Then, we rewrite the function $\nabla_\theta \mathcal{L}_{\mathrm{rdc}}(\theta_t)^T(\theta_t - \theta^*)$ as follows:

$$\nabla_\theta \mathcal{L}_{\mathrm{rdc}}(\theta_t)^T(\theta_t - \theta^*) = \nabla_\theta \mathcal{L}_{\mathrm{rdc}}(\theta_t)^T(\theta_t - \theta^*) - \nabla_\theta \mathcal{L}(\theta_t)^T(\theta_t - \theta^*) + \nabla_\theta \mathcal{L}(\theta_t)^T(\theta_t - \theta^*) \tag{36}$$

Combining the above equations we have:

$$\nabla_\theta \mathcal{L}_{\mathrm{rdc}}(\theta_t)^T(\theta_t - \theta^*) - \nabla_\theta \mathcal{L}(\theta_t)^T(\theta_t - \theta^*) + \nabla_\theta \mathcal{L}(\theta_t)^T(\theta_t - \theta^*) = \tag{37}$$

$$\frac{1}{2\alpha_t}\left(\|\alpha_t \nabla_\theta \mathcal{L}_{\mathrm{rdc}}(\theta_t)\|^2 + \|\theta_t - \theta^*\|^2 - \|\theta_{t+1} - \theta^*\|^2\right) \tag{38}$$

$$\nabla_\theta \mathcal{L}(\theta_t)^T(\theta_t - \theta^*) = \frac{1}{2\alpha_t}\left(\|\alpha_t \nabla_\theta \mathcal{L}_{\mathrm{rdc}}(\theta_t)\|^2 + \|\theta_t - \theta^*\|^2 - \|\theta_{t+1} - \theta^*\|^2\right) - \tag{39}$$

$$(\nabla_\theta \mathcal{L}_{\mathrm{rdc}}(\theta_t) - \nabla_\theta \mathcal{L}(\theta_t))^T(\theta_t - \theta^*) \tag{40}$$

Summing up the above equation for different value of $t \in [0, G-1]$ and the learning rate $\alpha_t$ is a constant $\alpha$, then we have:

$$\sum_{t=0}^{G-1} \nabla_\theta \mathcal{L}(\theta_t)^T(\theta_t - \theta^*) = \frac{1}{2\alpha}\|\theta_0 - \theta^*\|^2 - \|\theta_G - \theta^*\|^2 + \sum_{t=0}^{G-1}\left(\frac{1}{2\alpha}\|\alpha \nabla_\theta \mathcal{L}_{\mathrm{rdc}}(\theta_t)\|^2\right) \tag{41}$$

$$+ \sum_{t=0}^{G-1}\left((\nabla_\theta \mathcal{L}_{\mathrm{rdc}}(\theta_t) - \nabla_\theta \mathcal{L}(\theta_t))^T(\theta_t - \theta^*)\right) \tag{42}$$

Since $\|\theta_G - \theta^*\|^2 \geq 0$, we have:

$$\sum_{t=0}^{G-1} \nabla_\theta \mathcal{L}(\theta_t)^T(\theta_t - \theta^*) \leq \frac{1}{2\alpha}\|\theta_0 - \theta^*\|^2 + \sum_{t=0}^{G-1}\left(\frac{1}{2\alpha}\|\alpha \nabla_\theta \mathcal{L}_{\mathrm{rdc}}(\theta_t)\|^2\right) \tag{43}$$

$$+ \sum_{t=0}^{G-1}\left((\nabla_\theta \mathcal{L}_{\mathrm{rdc}}(\theta_t) - \nabla_\theta \mathcal{L}(\theta_t))^T(\theta_t - \theta^*)\right) \tag{44}$$

From the convexity of function $\mathcal{L}(\theta)$, we have:

$$\mathcal{L}(\theta_t) - \mathcal{L}(\theta^*) \leq \nabla_\theta \mathcal{L}(\theta_t)^T(\theta_t - \theta^*) \tag{45}$$

Combining the Equation 44 and Equation 45, we have:

$$\sum_{t=0}^{G-1} \mathcal{L}(\theta_t) - \mathcal{L}(\theta^*) \leq \frac{1}{2\alpha}\|\theta_0 - \theta^*\|^2 + \sum_{t=0}^{G-1} \left(\frac{1}{2\alpha}\|\alpha\nabla_\theta\mathcal{L}_{\texttt{rdc}}(\theta_t)\|^2\right) \tag{46}$$

$$+ \sum_{t=0}^{G-1} \left((\nabla_\theta\mathcal{L}_{\texttt{rdc}}(\theta_t) - \nabla_\theta\mathcal{L}(\theta_t))^T(\theta_t - \theta^*)\right) \tag{47}$$

We assume that $\|\theta - \theta^*\| \leq D$. Since $\|\nabla\mathcal{L}(\theta)\| \leq \sigma$, we have:

$$\sum_{t=0}^{G-1} \mathcal{L}(\theta_t) - \mathcal{L}(\theta^*) \leq \frac{D^2}{2\alpha} + \frac{G\alpha\sigma^2}{2} + \sum_{t=0}^{G-1} D(\|\nabla_\theta\mathcal{L}_{\texttt{rdc}}(\theta_t) - \nabla_\theta\mathcal{L}(\theta_t)\|) \tag{48}$$

Then:

$$\frac{\sum_{t=0}^{G-1}\mathcal{L}(\theta_t) - \mathcal{L}(\theta^*)}{G} \leq \frac{D^2}{2\alpha G} + \frac{\alpha\sigma^2}{2} + \sum_{t=0}^{G-1} \frac{D}{G}(\|\nabla_\theta\mathcal{L}_{\texttt{rdc}}(\theta_t) - \nabla_\theta\mathcal{L}(\theta_t)\|) \tag{49}$$

Since $\min(\mathcal{L}(\theta_t) - \mathcal{L}(\theta^*)) \leq \frac{\sum_{t=0}^{G-1}\mathcal{L}(\theta_t) - \mathcal{L}(\theta^*)}{G}$, we have:

$$\min(\mathcal{L}(\theta_t) - \mathcal{L}(\theta^*)) \leq \frac{D^2}{2\alpha G} + \frac{\alpha\sigma^2}{2} + \sum_{t=0}^{G-1} \frac{D}{G}(\|\nabla_\theta\mathcal{L}_{\texttt{rdc}}(\theta_t) - \nabla_\theta\mathcal{L}(\theta_t)\|) \tag{50}$$

We adopt $\varepsilon$ to denote $\|\nabla_\theta\mathcal{L}_{\texttt{rdc}}(\theta_t) - \nabla_\theta\mathcal{L}(\theta_t)\|$, then we have:

$$\min(\mathcal{L}(\theta_t) - \mathcal{L}(\theta^*)) \leq \frac{D^2}{2\alpha G} + \frac{\alpha\sigma^2}{2} + \sum_{t=0}^{G-1} \frac{D}{G}\varepsilon \tag{51}$$

$\square$

**Theorem A.1.** *The training loss on original dataset always monotonically decreases with every training epoch $t$, $\mathcal{L}(\theta_{t+1}) \leq \mathcal{L}(\theta_t)$ if it satisfies the condition that $\nabla_\theta\mathcal{L}(\theta_t)^T\nabla_\theta\mathcal{L}_{\texttt{rdc}}(\theta_t) \geq 0$ for $0 \leq t \leq G$ and the learning rate $\alpha \leq \min_t \frac{2}{L}\frac{\nabla_\theta\mathcal{L}(\theta_t)^T\nabla_\theta\mathcal{L}_{\texttt{rdc}}(\theta_t)}{\nabla_\theta\mathcal{L}_{\texttt{rdc}}(\theta_t)^T\nabla_\theta\mathcal{L}_{\texttt{rdc}}(\theta_t)}$.*

*Proof.* Since the training loss $\mathcal{L}(\theta)$ is lipschitz smooth, we have:

$$\mathcal{L}(\theta_{t+1}) \leq \mathcal{L}(\theta_t) + \nabla_\theta\mathcal{L}(\theta_t)^T\Delta\theta + \frac{L}{2}\|\Delta\theta\|^2, \tag{52}$$

$$\text{where} \qquad \Delta\theta = \theta_{t+1} - \theta_t. \tag{53}$$

Since, we are using SGD to optimize the reduced subset training loss $\mathcal{L}_{\texttt{rdc}}(\theta_t)$ model parameters. The update equation is:

$$\theta_{t+1} = \theta_t - \alpha\nabla_\theta\mathcal{L}_{\texttt{rdc}}(\theta_t) \tag{54}$$

Combining the above two equations, we have:

$$\mathcal{L}(\theta_{t+1}) \leq \mathcal{L}(\theta_t) + \nabla_\theta\mathcal{L}(\theta_t)^T(-\alpha\nabla_\theta\mathcal{L}_{\texttt{rdc}}(\theta_t)) + \frac{L}{2}\| - \alpha\nabla_\theta\mathcal{L}_{\texttt{rdc}}(\theta_t)\|^2 \tag{55}$$

Next, we have:

$$\mathcal{L}(\theta_{t+1}) - \mathcal{L}(\theta_t) \leq \nabla_\theta \mathcal{L}(\theta_t)^T (-\alpha \nabla_\theta \mathcal{L}_{\mathrm{rdc}}(\theta_t)) + \frac{L}{2} \| - \alpha \nabla_\theta \mathcal{L}_{\mathrm{rdc}}(\theta_t) \|^2 \tag{56}$$

From the above equation, we have:

$$\mathcal{L}(\theta_{t+1}) \leq \mathcal{L}(\theta_t), \quad \text{if} \quad \nabla_\theta \mathcal{L}(\theta_t)^T \nabla_\theta \mathcal{L}_{\mathrm{rdc}}(\theta_t) - \frac{\alpha L}{2} \|\nabla_\theta \mathcal{L}_{\mathrm{rdc}}(\theta_t)\|^2 \geq 0 \tag{57}$$

Since $\|\nabla_\theta \mathcal{L}_{\mathrm{rdc}}(\theta_t)\|^2 \geq 0$, we will have the necessary condition $\nabla_\theta \mathcal{L}(\theta_t)^T \nabla_\theta \mathcal{L}_{\mathrm{rdc}}(\theta_t) \geq 0$. Next, we rewrite the above condition as follows:

$$\nabla_\theta \mathcal{L}(\theta_t)^T \nabla_\theta \mathcal{L}_{\mathrm{rdc}}(\theta_t) \geq \frac{\alpha L}{2} \|\nabla_\theta \mathcal{L}_{\mathrm{rdc}}(\theta_t)\|^2 \tag{58}$$

Therefore, the necessary condition for the learning rate $\alpha$ is:

$$\alpha \leq \frac{2}{L} \frac{\nabla_\theta \mathcal{L}(\theta_t)^T \nabla_\theta \mathcal{L}_{\mathrm{rdc}}(\theta_t)}{\nabla_\theta \mathcal{L}_{\mathrm{rdc}}(\theta_t)^T \nabla_\theta \mathcal{L}_{\mathrm{rdc}}(\theta_t)} \tag{59}$$

Since the above condition needs to be true for all values for $t$, we have the following conditions for the learning rate:

$$\alpha \leq \min_t \frac{2}{L} \frac{\nabla_\theta \mathcal{L}(\theta_t)^T \nabla_\theta \mathcal{L}_{\mathrm{rdc}}(\theta_t)}{\nabla_\theta \mathcal{L}_{\mathrm{rdc}}(\theta_t)^T \nabla_\theta \mathcal{L}_{\mathrm{rdc}}(\theta_t)} \tag{60}$$

$\square$

## B    ABLATION STUDY

To study the contribution of each component in our learning framework, we conduct the following ablation study. `Q Target`: We replace the empirical returns used to update Q functions with the standard target Q function in the TD loss function. `Single Round`: We set the number of data selection rounds to 1 and study the function of multi-round data selection. The experimental results in Figure 4 show that removing any of these two modules will worsen the performance of REDOR. In case like `walker2d-medium`, ablation `Single Round` even decrease the performance by over 80%, and ablation `Q Target` results in a 95% performance drop in `walker2d-expert`. Furthermore, we also find that in the `halfcheetah` tasks, the impact of removing the two modules is relatively small. This result can be attributable to the fact that this task has a limited state space, and we can directly apply OMP to the entire dataset and identify important and diverse data.

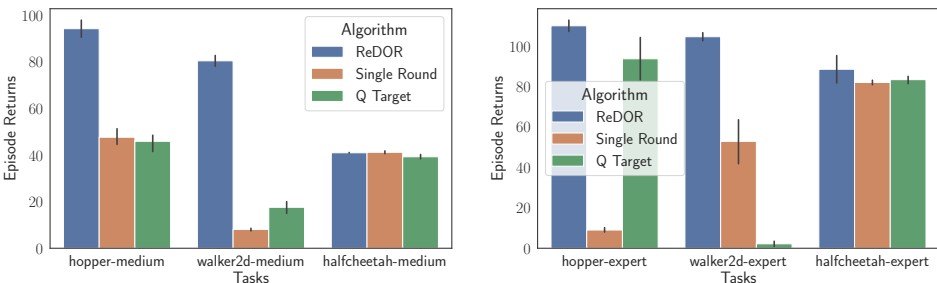

Figure 4: Ablation results on D4RL (Hard) tasks with the normalized score metric.

## C    COMPUTATIONAL COMPLEXITY

We report the computational overhead of REDOR on various datasets. All experiments are conducted on the same computational device (GeForce RTX 3090 GPU). The results in the following Table indicate that even on datasets containing millions of data points, the computational overhead remains low. This low computational complexity can be attributed to the trajectory-based selection technique in Sec. 4.2 (II) and the regularized constraint technique in Sec. 4.3, making our method easily scalable to large-scale datasets.

| Env | Data Number | REDOR |
|---|---|---|
| Hopper-medium-v0 | 999981 | 8m |
| Walker2d-medium-v0 | 999874 | 8m |
| Halfcheetah-medium-v0 | 998999 | 8m |
| Hopper-expert-v0 | 999034 | 8m |
| Walker2d-expert-v0 | 999304 | 8m |
| Halfcheetah-expert-v0 | 998999 | 8m |
| Hopper-medium-expert-v0 | 1199953 | 8m |
| Walker2d-medium-expert-v0 | 1999179 | 13m |
| Halfcheetah-medium-expert-v0 | 1997998 | 14m |
| Hopper-medium-replay-v0 | 200918 | 3m |
| Walker2d-medium-replay-v0 | 100929 | 3m |
| Halfcheetah-medium-replay-v0 | 100899 | 3m |

Table 4: The computational complexity associated with REDOR in various datasets. $m$ represents minutes.

## C.1 VISUALIZATION RESULTS

We visualize the selected data of ReDOR on various tasks based on the same method in Section 6.

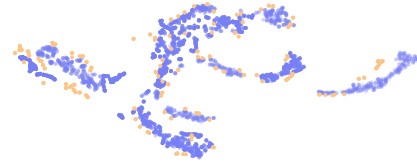

Figure 5: Visualization of selected data on hopper-medium-v0.

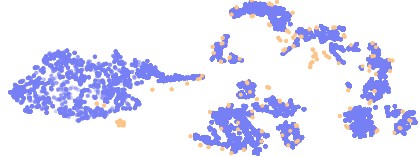

Figure 6: Visualization of selected data on hopper-medium-expert-v0.

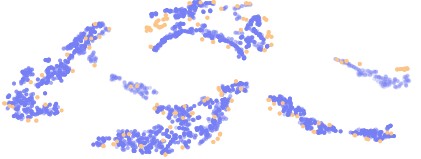

Figure 7: Visualization of selected data on hopper-expert-v0.

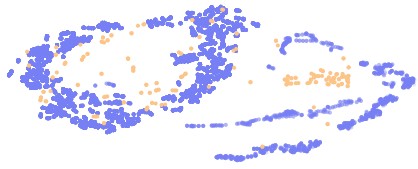

Figure 8: Visualization of selected data on walker2d-medium-expert-v0.

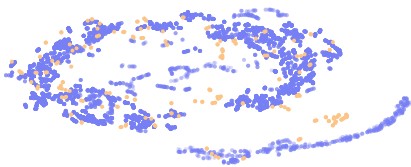

Figure 9: Visualization of selected data on walker2d-expert-v0.

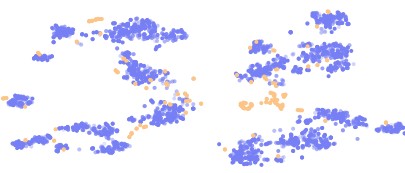

Figure 10: Visualization of selected data on halfcheetah-medium-v0.

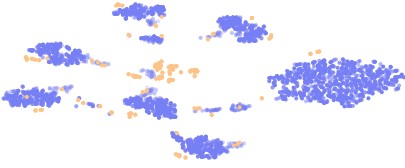

Figure 11: Visualization of selected data on halfcheetah-medium-expert-v0.

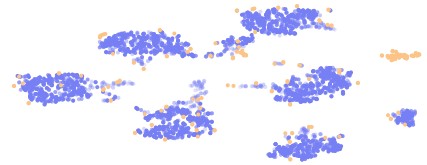

Figure 12: Visualization of selected data on halfcheetah-expert-v0.

# D  EXPERIMENTAL DETAILS

**Hyper-parameters.**  For the Mujoco tasks, we adopt the TD3+BC as the backbone of the offline algorithms. For the Antmaze tasks, we adopt the IQL as the backbone of the offline algorithms. We outline the hyper-parameters used by REDOR  in Table 5.

| Hyperparameter | Value |
|---|---|
| Optimizer | Adam |
| Critic learning rate | 3e-4 |
| Actor learning rate | 3e-4 |
| Mini-batch size | 256 |
| Discount factor | 0.99 |
| Target update rate | 5e-3 |
| Policy noise | 0.2 |
| Policy noise clipping | (-0.5, 0.5) |
| TD3+BC regularized parameter | 2.5 |
| Architecture | Value |
| Critic hidden dim | 256 |
| Critic hidden layers | 2 |
| Critic activation function | ReLU |
| Actor hidden dim | 256 |
| Actor hidden layers | 2 |
| Actor activation function | ReLU |
| REDOR Parameters | Value |
| Training rounds $T$ | 50 |
| $m$ | 50 |
| $\epsilon$ | 0.01 |

Table 5: Hyper-parameters sheet of  REDOR

