# OpenReview forum: "Fewer May Be Better: Enhancing Offline Reinforcement Learning with Reduced Dataset"
_ICLR.cc/2025/Conference — ICLR 2025 Poster_

### Official Review · Reviewer_6MgE · 2024-10-22

**Soundness:** 2
**Presentation:** 3
**Contribution:** 3
**Rating:** 6
**Confidence:** 4

**Summary:**

The authors introduce an approach for reducing the size of a dataset for offline RL by defining this reduction as a submodular set cover problem and using orthogonal matching pursuit. The resulting algorithm is evaluated on a modified version of D4RL locomotion tasks and the original antmaze tasks.

**Strengths:**

- This is an interesting and novel approach for data selection in RL. The high-level approach/formulation of the problem may be useful as a foundation for extensions.
- Strong results on a modified version of D4RL, and the unmodified antmaze.

**Weaknesses:**

There is a discrepancy between the proposed objectives and the resulting objectives that makes me question where the effectiveness of the proposed approach comes from.

The problem is initially defined as finding a subset of the data which results in a higher performing policy than the policy determined by training on the original dataset (Eqn 3). However, this is immediately discarded for another optimization problem, which instead tries to limit change in the value function (Eqn 5). While discovering a smaller dataset which achieves the same performance as the original dataset is an interesting problem, the authors claim in several places (and demonstrate) that their reduced dataset actually improves the performance. So where does the performance gain come from?

One possible cause for the performance increase is how the evaluation is done (add noisy/low performing trajectories to the D4RL dataset) and the filtering of low performing trajectories (Eqn 14). I would be very curious if this filtering alone is sufficient to also recover the performance of the algorithm. This concern, along with some missing key experimental details, makes me cautious about the experimental claims made in the paper.

Missing References which also filter the dataset using returns:
- [1] Chen, Xinyue, et al. "Bail: Best-action imitation learning for batch deep reinforcement learning." Advances in Neural Information Processing Systems 33 (2020): 18353-18363.
- [2] Yue, Yang, et al. "Boosting offline reinforcement learning via data rebalancing." arXiv preprint arXiv:2210.09241 (2022).

**Questions:**

Additional experiments:
- Does simply filtering the dataset by high returns recover the same performance?
- What is the performance of ReDOR on the original version of D4RL? One might expect that reducing mixed quality datasets like medium-expert, or medium, could also result in a high performance.

Missing experimental details:
- How is the hard dataset generated? How many datapoints are added to the dataset?
- How many datapoints are removed by ReDOR? What is the size of the reduced datasets?

General:
- Is there a way to tune the resulting dataset size?
- Is Fig 3, episode return = 99.5 for behaviors [2-7] correct or a bug?

---

> ### Author Response · Authors · 2024-11-22
> **Response to Reviewer 6MgE**
>
> Dear Reviewer,
>
> Thanks for finding our paper interesting and novel.
> We hope the following statement clear your concern.
>
> **W1 and Q1: Where does the performance gain come from? and Does simply filtering the dataset by high returns recover the same performance?**
>
> **A for W1 and Q1:**
> The reason for using Equation 5 instead of Equation 3 is that in the offline RL setting, we cannot directly solve the problem defined by Equation 3.
> For this reason, we attempt to adopt Equation 5 as an approximate alternative optimization problem.
>
> The performance gain comes from two aspects:
> (1) Reduced dataset can eliminate a large amount of redundant data, making the learning process of the algorithm more efficient.
> (2) we balance data quantity with performance by  focusing on data points that are aligned with the learned policy.
>
> As suggested, we conduct ablation study in the D4RL (Hard) tasks by simply filtering the dataset by high returns.
> The experimental results in Table 1 show that simply filtering data with high returns is not sufficient to achieve good performance.
> This is because the task cannot be well-defined if only given the expert demos.
> Diverse trajectories can help specify the boundary of the task.
>
> Finally, we have added the missing references as suggested in the revised version.
>
> | D4RL (Hard) | Simply Filtering | ReDOR |
> |:-:|:-:|:-:|
> |walker2d-medium-v0| 65.4$\pm$3.6 | 80.5$\pm$2.9 |
> |halfcheetah-medium-v0| 30.5$\pm$2.4 | 41.0$\pm$0.2 |
> |hopper-medium-v0| 91.2$\pm$3.2 | 94.3$\pm$4.6 |
> |walker2d-medium-replay-v0| 13.4$\pm$1.6 | 21.1$\pm$1.8 |
> |halfcheetah-medium-replay-v0| 40.2$\pm$0.3 | 41.1$\pm$0.4 |
> |hopper-medium-replay-v0| 31.0$\pm$3.1 | 35.3$\pm$3.2 |
> |walker2d-expert-v0| 86.4$\pm$2.6 | 104.6$\pm$2.5 |
> |halfcheetah-expert-v0| 87.2$\pm$5.3 | 88.5$\pm$2.4 |
> |hopper-expert-v0| 110.3$\pm$0.1 | 110.0$\pm$0.5 |
>
> Table 1. Comparison with the simply filtering baseline in the D4RL (Hard) tasks.
>
> **Q2: What is the performance of ReDOR on the original version of D4RL?**
>
> **A for Q2:**
> As suggested, we conduct additional experiments on the D4RL (Original) tasks.
> The experimental results in Table 2 show that the reduced dataset on the original dataset can still bring performance gains to the algorithm.
>
> | D4RL (Original) | Complete Dataset | ReDOR |
> |:-:|:-:|:-:|
> |walker2d-medium-v0| 79.7$\pm$1.8 | 89.3$\pm$2.3 |
> |halfcheetah-medium-v0| 42.8$\pm$0.3 | 45.2$\pm$0.2 |
> |hopper-medium-v0| 99.5$\pm$1.0 | 101.4$\pm$2.1 |
> |walker2d-medium-replay-v0| 25.2$\pm$5.1 | 40.1$\pm$3.8 |
> |halfcheetah-medium-replay-v0| 43.3$\pm$0.5 | 60.1$\pm$0.4 |
> |hopper-medium-replay-v0| 31.4$\pm$3.0 | 53.3$\pm$2.2 |
> |walker2d-expert-v0| 105.7$\pm$2.7 | 108.6$\pm$2.3 |
> |halfcheetah-expert-v0| 105.7$\pm$1.9 | 110.5$\pm$2.5 |
> |hopper-expert-v0| 112.2$\pm$0.2 | 115.0$\pm$0.5 |
>
> Table 2. Performance on the D4RL (Original) tasks.
>
> **Q3: How is the hard dataset generated? How many datapoints are added to the dataset?**
>
> **A for Q3:**
> We added noise data generated by various behavioral policies to the original dataset to simulate the noise in data collection in the real world.
> In each dataset, the size of the added noise data is 20\% of the original dataset.
>
> **Q4: How many datapoints are removed by ReDOR? What is the size of the reduced datasets?**
>
> **A for Q4:**
> In our experiments, each dataset reduces by approximately 70\% to 90\%.
> Specifically, for Walker2d and Hopper, there will be a reduction of 70\%, while for Halfcheetah, there will be a reduction of 90\%.
>
> **Q5: Is there a way to tune the resulting dataset size?**
>
> **A for Q5:**
> Yes, we can tune the size of the data subset by changing the approximation error $\epsilon$.
> For example, as $\epsilon$ increases, the precision of gradient matching decreases, resulting in a smaller data subset.
> Conversely, as $\epsilon$ decreases, the data subset becomes larger.
>
> **Q6: Is Fig 3, episode return = 99.5 for behaviors [2-7] correct or a bug?**
>
> **A for Q6:**
> Thanks for your suggestions.
> It is a bug and we have corrected it in the revised version.
>
> Thanks again for the valuable comments. We hope our response has cleared your concern. We are looking forward to more discussions

---

> > ### Comment · Reviewer_6MgE · 2024-11-24
> >
> > Thank you for the response, and the additional experiments.
> >
> > 1. The missing details you have provided here are helpful for understanding and reproducing your work. Can you make sure these appear in a revised draft of the paper?
> > 2. I'm not sure I agree that "simply filtering data with high returns is not sufficient to achieve good performance" based on the results you have provided in Table 1, since the performance looks very competitive.

---

> > > ### Author Response · Authors · 2024-11-25
> > > **Further response to Reviewer 6MgE**
> > >
> > > Dear Reviewer,
> > >
> > > Thanks for the quick reply. We will address your follow-up questions below.
> > >
> > > **1: Can the missing details appear in a revised draft of the paper?**
> > >
> > > **A1:**
> > > Yes, we assure that the added details and experiments will be incorporated into the revised version to enhance the clarity and persuasiveness of the paper.
> > > Once again, we express our gratitude to the reviewers.
> > > Your suggestions have greatly improved this manuscript.
> > >
> > > **2: Experiments of simply filtering data.**
> > >
> > > **A2:**
> > > This is because Mujoco tasks are relatively simple, so the performance looks very competitive.
> > > For this reason, we conduct additional experiments on more complex tasks, such as Adroit.
> > > Specifically, the dataset for Adroit tasks, including human and cloned data, is more realistic and collected by people.
> > > The experimental results in Table 1 show that in the more challenging dataset, ReDOR can achieve a significant performance improvement compared to the Simply Filtering baseline.
> > >
> > > | Adroit | Simply Filtering | ReDOR |
> > > |:-:|:-:|:-:|
> > > |pen-human-v0| 73.2$\pm$4.1 | 107.5$\pm$3.4 |
> > > |hammer-human-v0| 2.6$\pm$0.2 | 15.3$\pm$0.9 |
> > > |door-human-v0| 5.8$\pm$0.1 | 11.9$\pm$2.6 |
> > > |relocate-human-v0| 0.1$\pm$0.0 | 4.5$\pm$1.1 |
> > > |pen-cloned-v0| 40.6$\pm$2.9 | 103.4$\pm$4.4 |
> > > |hammer-cloned-v0| 3.7$\pm$0.7 | 24.2$\pm$7.0 |
> > > |door-cloned-v0| 2.4$\pm$0.3 | 12.4$\pm$2.7 |
> > > |relocate-cloned-v0| 0.3$\pm$0.1 | 2.4$\pm$0.3 |
> > >
> > > Table 1. Comparison with the simply filtering baseline in the Adroit tasks.
> > >
> > > Best,
> > >
> > > The Authors

---

> > > > ### Comment · Reviewer_6MgE · 2024-11-25
> > > >
> > > > Thanks for the additional experiments. I'm not convinced by the significance of the results, but it is clear that ReDOR does more than just filtering. I've updated my score (5 -> 6) to reflect this.

---

> > > > > ### Author Response · Authors · 2024-11-26
> > > > > **Thanks for raising the score to 6!**
> > > > >
> > > > > We would like to thank the reviewer for raising the score to 6! We also appreciate the valuable comments, which helped us significantly improve the paper's strengths.

---

### Official Review · Reviewer_wKBZ · 2024-10-24

**Soundness:** 3
**Presentation:** 3
**Contribution:** 3
**Rating:** 8
**Confidence:** 3

**Summary:**

The paper explores the interesting concept of finding a subset of the offline dataset to improve the performance of offline RL algorithms using orthogonal matching pursuit. The authors provide empirical and theoretical evidence of performance improvement on benchmark datasets.

**Strengths:**

1. The paper is well written and the idea is easy to follow.
2. The idea of subset selection is novel and interesting.
3. The paper provides both strong theoretical study and empirical analysis of the proposed method.

**Weaknesses:**

1. The authors characterize the field of offline RL only in terms of OOD action penalization and constraints on the behavior policy. There should also be a short discussion on model-based methods like MOPO [1] and MoERL [2], as some of these approaches have been shown to outperform model-free methods.

2. Some parts of the paper are difficult to understand without prior knowledge of orthogonal matching pursuit. Specifically, how is $F\lambda(s) = L_{max} - min_w Err_{\lambda} (w, S, L, \theta)$ used in the OMP.

3. If I understand correctly this method may not lead to the claimed reduction in complexity, as training $Q_{\theta}$ and $\pi_{\phi}$ till requires the full dataset.

Minor

The table references do not match the table numbers. On line 420, I believe the authors are referring to Table 1 instead of 6.2.

Suggestion : If the authors could include a notations table in Appendix it will help in readability and understanding the proofs.

References:
[1] Kidambi, Rahul, et al. "Morel: Model-based offline reinforcement learning." Advances in neural information processing systems 33 (2020): 21810-21823.
[2] Yu, Tianhe, et al. "Mopo: Model-based offline policy optimization." Advances in Neural Information Processing Systems 33 (2020): 14129-14142.

**Questions:**

Q1. How is the weight $w_i$ or $\lambda$ decided during training and the parameters $L_{max}$, $m$,$\epsilon$ chosen in practice?

Q2. Are the networks Qθ, πϕ networks first trained on the full dataset before starting with the subset selection?

Q3. What is the empirical reduction percentage achieved in each dataset?

Q4. In Figure 1 for the walker2d-expert-v0 environment, the reward first increases and then drops. It is also counterintuitive that the subset selected in ReDOR would perform better than a dataset containing only expert trajectories. Could the authors provide an explanation for this behavior?

Q5. Q5. Could the authors elaborate more on the Prioritize baseline, what do samples with highest TD Loss mean?

Q6. How does ReDOR perform on random datasets such as halfcheetah-random-v2?

Q7. I could not understand Fig 3. Why are the reduced dataset points more for category 6 when it is a subset of complete dataset?

---

> ### Author Response · Authors · 2024-11-22
> **Response to Reviewer wKBZ**
>
> Dear Reviewer,
>
> Thanks for finding our paper novel, interesting, strong theoretical study, and empirical analysis. We hope the following statement clear your concern.
>
> **W1: Short discussion on model-based methods like MOPO and MoERL.**
>
> **A for W1:**
> Thanks for your suggestions, we have corrected it in the revised version.
>
> **W2: How is $F_{\lambda}(s)=L_{\rm max}- min_wErr_{\lambda}(w, S, L, \theta)$ used in the OMP.**
>
> **A for W2:**
> Since $L_{\rm max}$ is a constant, maximizing $F_{\lambda}(s)$ is equivalent to minimizing $Err_{\lambda}(w, S, L, \theta)$.
> Therefore, we adopt OMP to directly minimize $Err_{\lambda}(w, S, L, \theta)$.
>
> **W3 and Q2: If the method leads to the claimed reduction in complexity and Are the networks $Q_{\theta}$, $\pi_{\phi}$ networks first trained on the full dataset before starting with the subset selection?**
>
> **A for W3 and Q2:**
> Yes, the selected data subset reduces the computational complexity.
> Specifically, we first train $Q_{\theta}$ and $\pi_{\phi}$ on the full dataset before starting with the subset selection.
> Then, we load the pre-trained parameter $\theta, \phi$ to select the data subset and re-train offline RL methods on the reduced dataset from scratch.
> The experimental results on the paper show that reduced dataset speeds up the training process.
>
> **Minor:**
>
> - **The table references do not match the table numbers:**
> Thanks for your suggestions, we have corrected it in the revised version.
>
> - **Notations table in Appendix:**
> Thanks for your suggestions, we have added the notation table in the revised version.
>
> **Q1: How is the weight $w_i$ or $\lambda$ decided during training and the parameters $L_{\rm max}, m, \epsilon$ chosen in practice?**
>
> **A for Q1:**
> In practice, $w_i$ and the indices $i$ are generated simultaneously by OMP.
> As for other parameters, we use uniform values across different tasks, $\lambda=0.1, m=50, \epsilon=0.01$.
> Since we use the OMP method to directly minimize $Err_{\lambda}(w, S, L, \theta)$, we do not need to set $L_{\rm max}$.
>
> **Q3: What is the empirical reduction percentage achieved in each dataset?**
>
> **A for Q3:**
> In our experiments, each dataset reduces by approximately 70\% to 90\%.
> Specifically, for Walker2d, there will be a reduction of 70\%, while for Halfcheetah, there will be a reduction of 90\%.
>
> **Q4: Why ReDOR performs better than a dataset containing only expert trajectories in Figure 1.**
>
> **A for Q4:**
> In the tasks depicted in Figure 1, we evaluate baselines on D4RL (Hard), which includes some suboptimal noisy data.
> We find that suboptimal data may lead to significant performance degradation due to distribution shifts.
> For this reason, ReDOR performs better than other baselines in the walker2d-expert-v0 task.
>
> **Q5: What do samples with highest TD Loss mean?**
>
> **A for Q5:**
> The $\texttt{Prioritize}$ baseline is designed based on the Prioritized Experience Replay for online RL [1].
> Specifically, when the value function is updated with gradients, prioritizing data with larger loss values provides more information for policy learning.
> Inspired by this, in each round of data selection, we choose data with highest TD Loss as the data subset.
>
> **Q6: How does ReDOR perform on random datasets such as halfcheetah-random-v2?**
>
> **A for Q6:**
> As suggested, we conduct experiments on the random dataset, as shown in the Table 1.
> The experimental results show that the performance improvement on the random dataset is not significant, which is because there is a relatively small amount of high-quality data in the random dataset.
>
> | | Complete Dataset | ReDOR |
> |:-:|:-:|:-:|
> |walker2d-random-v0| 1.4$\pm$1.6 | 2.4$\pm$1.0 |
> |halfcheetah-random-v0| 10.2$\pm$1.3 | 12.2$\pm$1.8 |
> |hopper-random-v0| 11.0$\pm$0.1 | 10.2$\pm$0.3 |
>
> Table 1. Experimental results on the random dataset.
>
> **Q7: Why are the reduced dataset points more for category 6 when it is a subset of complete dataset?**
>
> **A for Q7:**
> In Figure 3, each component reflects a distinct skill of the agent.
> The category 6 represents the stepping, which is relatively important compared with other skills, hence more points are allocated.
>
> We sincerely thank the reviewer again for the timely and valuable comments.
> We hope that our response and additional experimental results have cleared most of your concerns.
>
> [1] Tom Schaul, John Quan, Ioannis Antonoglou, and David Silver. Prioritized experience replay. arXiv
> preprint arXiv:1511.05952, 2015.

---

> > ### Comment · Reviewer_wKBZ · 2024-11-25
> > **Acknowledgement of Rebuttal**
> >
> > Thank you for the response. They have addressed my concerns. I am happy to recommend accept for this paper.

---

> > > ### Author Response · Authors · 2024-11-25
> > > **Thanks for raising the score to 8!**
> > >
> > > We would like to thank the reviewer for raising the score to 8! We also appreciate the valuable comments, which helped us significantly improve the paper's strengths.

---

### Official Review · Reviewer_W8YX · 2024-10-24

**Soundness:** 2
**Presentation:** 2
**Contribution:** 2
**Rating:** 5
**Confidence:** 4

**Summary:**

This paper introduces a method for dataset selection in offline reinforcement learning (RL) using the Orthogonal Matching Pursuit (OMP) algorithm and Monte Carlo Q losses. The proposed approach selects full trajectories whose loss gradients align well with the residuals.

**Strengths:**

1. The method demonstrates improved performance compared to the baselines.
2. The paper includes a theoretical analysis that provides a solid grounding for the approach.

**Weaknesses:**

1. Some key elements of the proposed algorithm are either missing or unclear, and there are some discrepancies between the paper and the accompanying codebase. For instance, the method used to generate "hard" datasets is not fully discussed in the paper, and the percentile $m$ mentioned in the paper differs from that in the codebase. More details are provided in the questions below.

2. Certain parts of the proposed algorithm may contain logical errors or inconsistencies. For example, in Line 4 of Algorithm 2, $r$ is a scalar, yet an inner product operation is applied to it. More details are provided in the questions below.

3. The baselines chosen for comparison seem somewhat outdated, which could affect the perceived significance of the performance improvements demonstrated by the proposed method.

**Questions:**

1. Could you please clarify how the suboptimal datasets for MuJoCo, namely "hard", were generated? The paper mentioned that they were generated by adding low-quality data, but the quality or source of such data and mix ratio should also be introduced.

2. Regarding $Q_\theta$ in Algorithm 1, could you explain how $Q_{\theta_t}$ was formulated? There is no update term in either the pseudocode or the codebase. Was $Q_\theta$ pretrained or trained simultaneously but omitted? It would be best if the pseudocode or thorough explanations were provided.

3. In Equation 14, it is stated that trajectories in the top $m%$ based on return are filtered, with $m$ set to 50, which would seem to exclude almost the entire random dataset. Could you provide the result of simply selecting trajectories with top $m (=50)\%$ returns for comparison?

4. In the codebase, it seems that in addition to the evaluation of Monte Carlo Q targets, the selection of candidate trajectories via OMP is filtered based on trajectory returns. What is the exact search space of the selected trajectories? If it is the filtered one with trajectory returns, then how can we ensure the fairness of the comparison to baselines that do not utilize such a filter?

5. In the paper, the percentile $m$ is specified as 50 (Top), but in the codebase, it varies (Bottom 50, 70, and 95). Could you clarify the reason for this difference?

6. In Algorithm 2, $r$ is defined as a scalar, but in Line 4, an inner product is applied. Could you kindly explain this?

7. In Line 3 of Algorithm 2, the inequality appears to be reversed. Is this correct?

8. Is there a reason why TD3+BC was chosen as the backbone offline RL algorithm for the MuJoCo tasks? Would using IQL, as in the Antmaze tasks, provide a more consistent comparison?

9. For the MuJoCo tasks, the authors used the "-v0" versions, which are now outdated and differ from the more recent "-v2" versions. Could you explain the reasoning behind using "-v0"?

10. For the "Complete Dataset" scores in the Antmaze tasks, it seems that these values are taken from the IQL paper, which does not provide standard deviations. Could you clarify how these scores were derived?

11. While the baselines used in the experiments appear somewhat dated, dataset selection has recently gained increased attention in offline RL. Hence, it seems that recent algorithms should be contained as baselines. For example, "Improving Generalization in Offline Reinforcement Learning via Adversarial Data Splitting (Wang et al., 2024)" provides a codebase, which could allow for a straightforward comparison. Or, is there any reason why such comparisons are inappropriate?

12. Could you provide more details on what is meant by the "Complete Dataset" baseline? Specifically, is it the original mixture of the desired dataset and the suboptimal dataset, or is it just the original dataset?

---

> ### Author Response · Authors · 2024-11-22
> **Response to Reviewer W8YX (I)**
>
> Dear Reviewer,
>
> Thanks for your valuable comments. We hope the following statement can address your concern.
>
> **W1 and Q1: How to generate hard datasets.**
>
> **A for W1 and Q1:**
> We added noise data generated by various behavioral policies to the original dataset to simulate the noise in data collection in the real world.
> In each dataset, the size of the added noise data is 20\% of the original dataset.
>
> **Q2: How was $Q_{\theta}$ formulated?**
>
> **A for Q2:**
> We first train $Q_{\theta}$ on the full dataset before starting with the subset selection.
> Then, we load the pre-trained parameter $\theta$ to select the data subset and re-train offline RL methods on the reduced dataset from scratch.
>
> **Q3: The results of simply selecting trajectories with top returns for comparison.**
>
> **A for Q3:**
> As suggested, we conduct ablation study in the D4RL (Hard) tasks by simply filtering the dataset by high returns ($m=50$).
> The experimental results in Table 1 show that simply filtering data with high returns is not sufficient to achieve good performance.
> This is because the task cannot be well-defined if only given the expert demos.
> Diverse trajectories can help specify the boundary of the task.
>
> | D4RL (Hard) | Simply Filtering | ReDOR |
> |:-:|:-:|:-:|
> |walker2d-medium-v0| 65.4$\pm$3.6 | 80.5$\pm$2.9 |
> |halfcheetah-medium-v0| 30.5$\pm$2.4 | 41.0$\pm$0.2 |
> |hopper-medium-v0| 91.2$\pm$3.2 | 94.3$\pm$4.6 |
> |walker2d-medium-replay-v0| 13.4$\pm$1.6 | 21.1$\pm$1.8 |
> |halfcheetah-medium-replay-v0| 40.2$\pm$0.3 | 41.1$\pm$0.4 |
> |hopper-medium-replay-v0| 31.0$\pm$3.1 | 35.3$\pm$3.2 |
> |walker2d-expert-v0| 86.4$\pm$2.6 | 104.6$\pm$2.5 |
> |halfcheetah-expert-v0| 87.2$\pm$5.3 | 88.5$\pm$2.4 |
> |hopper-expert-v0| 110.3$\pm$0.1 | 110.0$\pm$0.5 |
>
> Table 1. Comparison with the simply filtering baseline in the D4RL (Hard) tasks.
>
> **Q4: What is the exact search space of the selected trajectories?**
>
> **A for Q4:**
> To balance the data quantity with performance, we aim to focus on data points that are aligned with the learned policy, avoiding performance degradation caused by suboptimal trajectories.
> As suggested, we conduct additional experiments by adding the same filter module in the baselines.
> The experimental results in Table 2 show that ReDOR still performs better than baselines equipped with the filter module.
> This is because the reduced dataset can eliminate a large amount of redundant data, making the learning process of the algorithm more efficient.
>
> | D4RL (Hard) | Random (Filter) | Prioritized (Filter) | ReDOR |
> |:-:|:-:|:-:|:-:|
> |walker2d-medium-v0| 52.4$\pm$2.7 | 60.6$\pm$4.5 | 80.5$\pm$2.9 |
> |halfcheetah-medium-v0| 25.7$\pm$1.2 | 32.4$\pm$0.4 | 41.0$\pm$0.2 |
> |hopper-medium-v0| 90.7$\pm$3.9 | 92.2$\pm$2.7 | 94.3$\pm$4.6 |
> |walker2d-medium-replay-v0| 11.4$\pm$2.4 | 16.3$\pm$1.4 | 21.1$\pm$1.8 |
> |halfcheetah-medium-replay-v0| 27.2$\pm$0.6 | 28.9$\pm$0.8 | 41.1$\pm$0.4 |
> |hopper-medium-replay-v0| 22.6$\pm$2.5 | 34.4$\pm$1.7 | 35.3$\pm$3.2 |
> |walker2d-expert-v0| 79.3$\pm$3.2 | 89.7$\pm$2.9 | 104.6$\pm$2.5 |
> |halfcheetah-expert-v0| 79.6$\pm$1.9 | 68.1$\pm$3.3 | 88.5$\pm$2.4 |
> |hopper-expert-v0| 108.4$\pm$0.8 | 109.4$\pm$0.9 | 110.0$\pm$0.5 |
>
> Table 2. Comparison with baselines in the D4RL (Hard) tasks.
>
> **Q5 and W1: The selection of $m$.**
>
> **A for Q5 and W1:**
> We thank the reviewer for raising the point.
> Due to poor communication among collaborators, the initial codebase was inadvertently uploaded.
>
> **Q6 and W2: The inner product operation in Algorithm 2.**
>
> **A for Q6 and W2:**
> $r$ is not scalar.
> In practice, we use the last layer of the gradients for neural networks and $r$ denotes the residual error, which is a vector and the shape is ($\cdot$,257).
>
> **Q7: The inequality in Algorithm 2 appears to be reversed.**
>
> **A for Q7:**
> Thanks for your suggestions, we have corrected it in the revised version.
>
> **Q8: Is there a reason why TD3+BC was chosen as the backbone offline RL algorithm for the MuJoCo tasks? and Would using IQL, as in the Antmaze tasks, provide a more consistent comparison?**
>
> **A for Q8:**
> TD3+BC algorithm is one of the well-known algorithms in the offline RL community, and we have chosen TD3+BC as the backbone.
> As suggested, we conduct additional experiments in MuJoCo tasks based on IQL.
> The experimental results in Table 3 show that IQL can still be adapted to Mujoco tasks with ReDOR.
>
> | D4RL (Hard) | ReDOR (IQL) | ReDOR |
> |:-:|:-:|:-:|
> |walker2d-medium-v0| 71.3$\pm$2.1 | 80.5$\pm$2.9 |
> |halfcheetah-medium-v0| 40.9$\pm$0.3 | 41.0$\pm$0.2 |
> |hopper-medium-v0| 96.4$\pm$2.3 | 94.3$\pm$4.6 |
> |walker2d-medium-replay-v0| 23.6$\pm$3.6 | 21.1$\pm$1.8 |
> |halfcheetah-medium-replay-v0| 43.1$\pm$0.9 | 41.1$\pm$0.4 |
> |hopper-medium-replay-v0| 39.1$\pm$4.4 | 35.3$\pm$3.2 |
> |walker2d-expert-v0| 106.6$\pm$1.4 | 104.6$\pm$2.5 |
> |halfcheetah-expert-v0| 108.2$\pm$2.7 | 88.5$\pm$2.4 |
> |hopper-expert-v0| 110.5$\pm$0.4 | 110.0$\pm$0.5 |
>
> Table 3. Experimental results of ReDOR (IQL) in the D4RL (Hard) tasks.

---

> ### Author Response · Authors · 2024-11-22
> **Response to Reviewer W8YX (II)**
>
> **Q9: Why use -v0 version?**
>
> **A for Q9:**
> TD3+BC algorithm in the original paper was evaluated on the -v0 version, so we also conduct experiments on the -v0 version, for no other reason.
>
> **Q10: The standard deviations of IQL.**
>
> **A for Q10:**
> We run the official code released by the IQL authors, supplementing the missing standard deviation in the paper.
>
> **Q11: Recent baselines in dataset selection of offline RL.**
>
> **A for Q11:**
> The baselines considered in this paper are common approach to data selection in supervised learning.
> This is also why we chose these methods as our baselines.
> As suggested, we evaluate the recent baselines (ADS [1]) in the D4RL (Hard) tasks.
> The experimental results in Table 4 show that ReDOR achieves better performance than ADS in most tasks.
> We thank the suggestions of the reviewer, and we will place the complete experimental results and discussion in the revised version later.
>
> | D4RL (Hard) | ADS | ReDOR |
> |:-:|:-:|:-:|
> |walker2d-medium-v0| 78.9$\pm$1.5 | 80.5$\pm$2.9 |
> |halfcheetah-medium-v0| 37.3$\pm$0.6 | 41.0$\pm$0.2 |
> |hopper-medium-v0| 91.4$\pm$2.8 | 94.3$\pm$4.6 |
> |walker2d-medium-replay-v0| 16.4$\pm$2.9 | 21.1$\pm$1.8 |
> |halfcheetah-medium-replay-v0| 40.4$\pm$0.6 | 41.1$\pm$0.4 |
> |hopper-medium-replay-v0| 31.9$\pm$2.7 | 35.3$\pm$3.2 |
> |walker2d-expert-v0| 98.9$\pm$2.8 | 104.6$\pm$2.5 |
> |halfcheetah-expert-v0| 88.2$\pm$1.3 | 88.5$\pm$2.4 |
> |hopper-expert-v0| 104.2$\pm$0.6 | 110.0$\pm$0.5 |
>
> Table 4. Comparison with ADS in the D4RL (Hard) tasks.
>
> **Q12: More details on what is meant by the "Complete Dataset" baseline.**
>
> **A for Q12:**
> In the original D4RL tasks, the complete dataset baseline denote the original dataset.
> As for the D4RL (Hard) tasks, the complete dataset denote the original mixture of the desired dataset and the suboptimal dataset.
>
> Thanks again for the valuable comments. We hope our additional experimental results and explanation have cleared the concern. We sincerely hope that the reviewer can re-evaluate our paper after seeing the our response. More comments on further improving the presentation are also very much welcomed.
>
> [1] Wang, Da, et al. "Improving Generalization in Offline Reinforcement Learning via Adversarial Data Splitting." Forty-first International Conference on Machine Learning.

---

> ### Author Response · Authors · 2024-11-25
> **Looking forward to further comments!**
>
> Dear Reviewer,
>
> We have conducted additional experiments on ablations and baselines. We are wondering if our response and revision have cleared your concerns. We would appreciate it if you could kindly let us know whether you have any other questions. We are looking forward to comments that can further improve our current manuscript. Thanks!
>
> Best regards,
>
> The Authors

---

> > ### Comment · Reviewer_W8YX · 2024-11-27
> >
> > Thank you for your clarification. Most concerns regarding implementation are addressed. I adjusted my score accordingly.

---

> > > ### Author Response · Authors · 2024-11-27
> > > **Thanks for raising the score!**
> > >
> > > We would like to thank the reviewer for raising the score! We also appreciate the valuable comments, which helped us significantly improve the paper's strengths.

---

### Official Review · Reviewer_uw5n · 2024-11-02

**Soundness:** 2
**Presentation:** 2
**Contribution:** 2
**Rating:** 5
**Confidence:** 3

**Summary:**

Motivated by the large size of the offline dataset as well as suboptimal data quality in offline RL, this paper considers the problem of finding a coreset out of the given dataset. The authors first formulate such problem as a task to approximate the actual gradients (from the complete dataset) in the offline training process. And a line of of results are provided to support the low approximation errors. Then the method named Reduced Datasets for Offline RL (REDOR) is proposed, inspired by the orthogonal matching pursuit (OMP). Finally, the method is compared with several baseline methods on D4RL data.

**Strengths:**

> **Originality**
- Such new method is proposed to select a coreset from the raw offline dataset, which could contribute as an alternative approach in offline RL.

> **Clarity**
- Several informative figures are provided. Especially the one by t-SNE provides a straightforward way to understand the behaviour of such selection process.

> **Significance**
- In some of the settings concerned in the experiments, such method is quite efficient.

**Weaknesses:**

> **Quality**
- Several assumptions in Theorem 4.1 are rather stronger than scenarios in actual implementations. One observation often seen in offline RL is the diverging gradients (if without proper training techniques), which, however, are assumed to be uniformly bounded in the paper, w.r.t parameters in respectively policies and Q-functions.
- Despite the multi-round selection strategy introduced in Section 4.2, as long as the empirical returns are used, as depicted in equation (13), the targets in training steps are relatively fixed (in the sense of distributions due to behaviour policies), which then makes (13) no longer an approximation of Bellman backup errors. As a result, it is currently not clear if such approach would lead to a guaranteed good estimation of values/Q-functions.
- According to what the reviewer can understand about the statements and proof for results in Section 5, the theorems only consider the proposed method defined with classic TD loss, while do not consider the techniques emphasized in Section 4.2 - 4.3. As a result, such theoretical discussion is not an actual analysis of the proposed algorithm (feel free to correct me).
- In Line 766, within the proof for Theorem 5.2, it is not justified why $S\^k$ can always start from the cluster center ${c\_k}$ of gradients.

> **Clarity**
- According to the way a Q-function is defined in Line 99, some index of $t$ should be included in the notation of $Q$.
- Horizon $H$ is not explicitly defined.
- There is not enough information for $L\_{\text{max}}$.
- There lacks for an introduction to how KRLS, Log-Det and BlockGreedy are implemented in such offline RL settings.

> **Significance**
- As explained in the 'Quality' part, the theoretical results seem not to be exactly for the proposed method.

**Questions:**

None

---

> ### Author Response · Authors · 2024-11-22
> **Response to Reviewer uw5n**
>
> Dear Reviewer,
>
> Thanks for your valuable and detailed comments.
> We hope the following statement clear your concern.
>
> **W1: The assumptions in Theorem 4.1 are rather stronger than scenarios in actual implementations.**
>
> **A for W1:**
> We thank the reviewer for raising the point.
> If the gradients of the algorithm diverge in practice, it would contradict our assumptions, and the selected data subset would no longer be valuable.
> However, there are currently various empirical techniques [1,2,3] that can overcome this issue, ensuring that the algorithm's gradients remain stable.
>
> **W2: If such approach would lead to a guaranteed good estimation of values/Q-functions.**
>
> **A for W2:**
> In practice, we found that if we use the standard Bellman backup errors, the gradients used for data selection can be unstable.
> On the other hand, if we use the relatively fixed target, it cannot lead to the good estimation of Q-functions.
> To address this issue, we first train $Q_{\theta}$ and $\pi_{\phi}$ on the full dataset based on the standard Bellman backup errors before starting with the subset selection.
> Then, we load the pre-trained parameter $\theta, \phi$ to select the data subset based on Equation 13, which can ensure relatively accurate and stable gradients.
>
> **W3: The theorems only consider the proposed method defined with classic TD loss.**
>
> **A for W3:**
> The theoretical analysis in Section 5 provides conclusions when the gradient approximation error is $\epsilon$.
> On the other hand, the techniques in Sections 4.2-4.3 can guarantee the gradient approximation error is lower than $\epsilon$.
> Therefore, the theoretical conclusion in Section 5 can be adapted into our algorithm.
>
> **W4: Why $S^k$ can always start from the cluster center of gradients.**
>
> **A for W4:**
> This is the assumption in our theoretical analysis.
> However, this assumption is not difficult to achieve in practice.
> We can first cluster the dataset and using the cluster center as the initial point before selecting the data subset within each cluster.
>
> **C1: Some index $t$ of should be included in the notation of $Q$.**
>
> **A for C1:**
> Thanks for your suggestions, we have corrected it in the revised version.
>
> **C2: Horizon $H$ is not explicitly defined.**
>
> **A for C2:**
> Thanks for your suggestions, we have corrected it in the revised version.
>
> **C3: There is not enough information for $L_{\rm max}$.**
>
> **A for C3:**
> Since $L_{\rm max}$ is a constant, maximizing $F_{\lambda}(s)$ is equivalent to minimizing $Err_{\lambda}(w, S, L, \theta)$.
> Therefore, we adopt OMP to directly minimize $Err_{\lambda}(w, S, L, \theta)$ and do not need to set $L_{\rm max}$.
>
> **C4: There lacks for an introduction to how KRLS, Log-Det and BlockGreedy are implemented in such offline RL settings.**
>
> **A for C4:**
> We implement KRLS, Log-Det and BlockGreedy based on the standard Bellman backup errors, and the specific data selection process remains consistent with the original paper.
>
> We sincerely thank the reviewer again for the timely and valuable comments. We hope that our response have cleared most of your concerns.
>
> [1] Aviral Kumar, Aurick Zhou, George Tucker, and Sergey Levine. Conservative q-learning for offline reinforcement learning. Advances in Neural Information Processing Systems, 33:1179–1191, 2020.
>
> [2] Scott Fujimoto and Shixiang Shane Gu. A minimalist approach to offline reinforcement learning. Advances in neural information processing systems, 34:20132–20145, 2021.
>
> [3] Ilya Kostrikov, Ashvin Nair, and Sergey Levine. Offline reinforcement learning with implicit q-learning. arXiv preprint arXiv:2110.06169, 2021.

---

> ### Author Response · Authors · 2024-11-25
> **Looking forward to further comments!**
>
> Dear Reviewer,
>
> We have added additional explanations for our methods. We are wondering if our response and revision have cleared your concerns. We would appreciate it if you could kindly let us know whether you have any other questions. We are looking forward to comments that can further improve our current manuscript. Thanks!
>
> Best regards,
>
> The Authors

---

> ### Comment · Reviewer_uw5n · 2024-11-25
>
> Thanks for the responses! Please see my follow-ups:
>
> > **If the gradients of the algorithm diverge in practice, it would contradict our assumptions, and the selected data subset would no longer be valuable. However, there are currently various empirical techniques [1,2,3] that can overcome this issue, ensuring that the algorithm's gradients remain stable.**
>
> - First of all, in the current proposed method it has not been clearly introduced how such boundedness is guaranteed. In addition, either empirically or theoretically, it would be better if one can verify those boundedness assumptions are satisfied. Otherwise such strong assumptions may restrict the capabilities of the method.
>
> > **A for W2**
>
> - So it means the target is actually biased? (not following the Bellman operator)?
>
> - If some pre-training on whole dataset is needed, then the advantage of acceleration could not hold?
>
> > **the techniques in Sections 4.2-4.3 can guarantee the gradient approximation error is lower than $\\epsilon$**
>
> - I see the techniques, but not the justification. For example, both theorems explicitly use the term 'TD loss', then why is it the case in Equation (14), that is not the TD loss?

---

> > ### Author Response · Authors · 2024-11-26
> > **Further response to Reviewer uw5n**
> >
> > Dear Reviewer,
> >
> > Thanks for your reply! We will address your follow-up questions below.
> >
> > **Q1: The boundedness of gradient in offline RL is not guaranteed.**
> >
> > **A for Q1:**
> > We agree with the comment of the Reviewer.
> > Current offline RL methods can only ensure that the Q-values do not diverge.
> > Although this can, to some extent, reflect the gradients of the Q-network that have not diverged, there is no rigorous proof that the bounds of the gradients can be guaranteed.
> > We appreciate the Reviewer pointing out this issue, and we have added a subsection in the revised paper (l316-l323), taking your suggestion as a limitation of the theoretical analysis, thereby providing valuable insights for future research.
> >
> > **Q2.1 and Q3: The target is actually biased, and why is it the case in Equation (14), that is not the TD loss?**
> >
> > **A for Q2.1 and Q3:**
> > Compared to updates in standard RL methods, the targets we use are indeed biased.
> > However, the reason we use Equation 14 instead of TD Loss is to provide a more consistent learning signal and mitigate instability caused by changing target values, thereby making the selected data more valuable.
> > We thank you again for your valuable comments, and we have highlighted this point in the discussion of limitations in our revised paper (l316-l323).
> > Moreover, we conduct additional ablation studies by replacing the empirical returns module in ReDOR with the standard TD Loss (ReDOR (TD Loss)).
> > The experimental results in Table 1 show that the empirical returns module is necessary.
> >
> > | D4RL (Hard) | ReDOR (TD Loss) | ReDOR |
> > |:-:|:-:|:-:|
> > |walker2d-medium-v0| 20.4$\pm$3.1 | 80.5$\pm$2.9 |
> > |halfcheetah-medium-v0| 38.4$\pm$0.3 | 41.0$\pm$0.2 |
> > |hopper-medium-v0| 42.1$\pm$2.3 | 94.3$\pm$4.6 |
> > |walker2d-medium-replay-v0| 11.4$\pm$1.2 | 21.1$\pm$1.8 |
> > |halfcheetah-medium-replay-v0| 20.2$\pm$0.3 | 41.1$\pm$0.4 |
> > |hopper-medium-replay-v0| 13.0$\pm$1.1 | 35.3$\pm$3.2 |
> > |walker2d-expert-v0| 3.4$\pm$0.2 | 104.6$\pm$2.5 |
> > |halfcheetah-expert-v0| 80.2$\pm$0.5 | 88.5$\pm$2.4 |
> > |hopper-expert-v0| 90.3$\pm$5.4 | 110.0$\pm$0.5 |
> >
> > Table 1. Ablation about the empirical returns module in the D4RL (Hard) tasks.
> >
> > **Q2.2: If some pre-training on whole dataset is needed, then the advantage of acceleration could not hold?**
> >
> > **A for Q2.2:**
> > Although pre-training is required on the whole dataset, when we need to retrain the algorithm (e.g., parameter tuning), we can significantly reduce our training time on the reduced dataset.
> >
> > Best,
> >
> > The Authors

---

> ### Comment · Reviewer_uw5n · 2024-11-28
>
> Thanks for the details.
>
> > **Boundedness issue of gradients**
>
> - I appreciate that the authors explicitly clarify such issue as a limitation in the corresponding part of the paper.
>
> > **Using TD loss or not**
>
> - The result attached looks interesting, roughly showing that consistency in such setting may not always be ideal. Just want to confirm, when adopting the TD loss version, i.e. the first column in the table, some pre-training similar as the primary procedure has also been used?
>
> > **Training cost**
>
> - It's ok not to have such results currently, but could help for some later revision if an empirical comparison between different approaches (with or without pre-training) is provided.

---

> > ### Author Response · Authors · 2024-12-01
> > **Further response to Reviewer uw5n**
> >
> > Dear Reviewer,
> >
> > Thank you for your thoughtful feedback on our paper. With only two days remaining in the discussion period, we kindly ask that you review our responses to ensure we have fully addressed your concerns. If you find our responses satisfactory, we would greatly appreciate it if you could reconsider your rating/scoring.
> >
> > Your engagement and constructive input have been invaluable, and we truly appreciate your time and effort in supporting this process.
> >
> > Best regards,
> >
> > Authors

---

> > > ### Comment · Reviewer_uw5n · 2024-12-02
> > >
> > > The reviewer appreciates the discussion and clarification. As highlighted in the comments, the current theoretical analysis relies on relatively strong assumptions, and the results do not yet fully align with the complete set of techniques proposed. With further investigation and refinement, the paper has the potential to provide deeper insights into these problem settings.

---

> ### Author Response · Authors · 2024-11-29
> **Further response to Reviewer uw5n**
>
> Dear Reviewer,
>
> Thanks for your reply! We hope the following statement can clear your remaining concerns.
>
> **Boundedness issue of gradients.**
>
> **A:**
> Thanks again for your suggestions.
> We also appreciate the valuable comments, which helped us significantly improve the paper's strengths.
>
> **Using TD loss or not.**
>
> **A:**
> Yes.
> In the TD loss version, all pre-training, similar to the primary procedure, has also been used besides using TD loss.
>
> **Training cost.**
>
> **A:**
> As suggested, we conduct additional experiments for the computational cost between different approaches (with or without pre-training).
> Specifically, we train TD3+BC and IQL on the reduced dataset generated by ReDOR (named TD3+BC (Reduce) and IQL (Reduce)).
> On the other hand, we train TD3+BC and IQL on the standard datasets (named TD3+BC (original) and IQL (original)).
> We record the time the algorithms reach the same performance.
> The experimental results in Table 1 show that the computational cost spent training on the reduced dataset is significantly lower than on the original dataset.
>
> Note that the results for IQL are generally faster than those for TD3+BC because we are using the official code for IQL, implemented with JAX, which is faster than PyTorch.
> All experiments are conducted on the GeForce RTX 3090 GPU device.
> In the revised version later, we will conduct an empirical comparison between more approaches.
>
> | D4RL | TD3+BC (Reduce) | TD3+BC (original) | IQL (Reduce) | IQL (original) |
> |:-:|:-:|:-:|:-:|:-:|
> |walker2d-medium-v0| 47m | 81m | 23m | 38m |
> |halfcheetah-medium-v0| 46m | 81m | 21m | 35m |
> |hopper-medium-v0| 33m | 67m | 22m | 36m |
> |walker2d-medium-replay-v0| 44m | 78m | 21m | 34m |
> |halfcheetah-medium-replay-v0| 27m | 49m | 23m | 37m |
> |hopper-medium-replay-v0| 25m | 45m | 21m | 35m |
> |walker2d-expert-v0| 49m | 87m | 20m | 32m |
> |halfcheetah-expert-v0| 56m | 93m | 26m | 41m |
> |hopper-expert-v0| 14m | 24m | 14m | 20m |
>
> Table 1. Computational cost between different approaches (with or without pre-training). m denotes the minute.
>
> Best,
>
> The Authors

---

### Meta-Review · Area_Chair_JKMe · 2024-12-22

**Metareview:**

The paper discusses subset selection for offline RL datasets to improve the performance of RL methods.
The reviewers think that the paper is well written and the idea is easy to follow. The paper present both theory and experiments for the proposed method.

In terms of weaknesses, there seems to be a gap between the proposed theory and the actual experiments, in the sense that the theoretical results seem not to be for the proposed method.
The baselines used for comparison are also a bit outdated.

**Additional Comments On Reviewer Discussion:**

Some of the reviewers concerns were addressed during the rebuttal.

---

### Decision · Program_Chairs · 2025-01-22

Accept (Poster)